



# Reconstructing Holocene temperatures in time and space using paleoclimate data assimilation

Michael P. Erb[1], Nicholas P. McKay[1], Nathan Steiger[2,3], Sylvia Dee[4], Chris Hancock[1], Ruza F. Ivanovic[5], Lauren J. Gregoire[5], and Paul Valdes[6]

[1]School of Earth and Sustainability, Northern Arizona University, Flagstaff, AZ, USA
[2]Lamont-Doherty Earth Observatory, Columbia University, New York, NY, USA
[3]Institute of Earth Sciences, Hebrew University, Jerusalem, Israel
[4]Department of Earth, Environmental, and Planetary Sciences, Rice University, Houston, TX, USA
[5]School of Earth and Environment, University of Leeds, Leeds, UK
[6]School of Geographical Sciences, University of Bristol, Bristol, UK

Correspondence to: Michael P. Erb (michael.erb@nau.edu)

**Abstract.** Paleoclimatic records provide valuable information about Holocene climate, revealing aspects of climate variability for a multitude of sites around the world. However, such data also possess limitations. Proxy networks are spatially uneven, seasonally biased, uncertain in time, and present a variety of challenges when used in concert to illustrate

the complex variations of past climate. Paleoclimatic data assimilation provides one approach to reconstructing past climate that can account for the diverse nature of proxy records while maintaining the physics-based covariance structures simulated by climate models. Here, we use paleoclimate data assimilation to create a spatially-complete reconstruction of temperature over the past 12,000 years using proxy data from the Temperature 12k database and output from transient climate model simulations. Following the last glacial period, the reconstruction shows Holocene temperatures warming to a peak near 6,400

years ago followed by a slow cooling toward the present day, supporting a preindustrial global mean surface temperature maximum during the mid-Holocene. Sensitivity tests show that if proxies have an overlooked summer bias, some apparent mid-Holocene warmth could actually represent summer trends rather than annual mean trends. Regardless, the potential effects of proxy seasonal biases are insufficient to align reconstructed global mean temperature with the warming trends seen in transient model simulations.

## 1. Introduction

Paleoclimate research is typically conducted in two ways: by extracting information from natural archives of past climate, called climate proxies, and by simulating past climate with models. These two methods have complementary strengths, as proxies provide location-specific data about climate, while models can be used to explore spatial and dynamical relationships in the broader climate system. Here, we use paleoclimate data assimilation to synthesize information from both

approaches into a spatially complete reconstruction of Holocene temperature (approximately the past 11,700 years).



Compared to the last deglaciation and contemporary global warming, the Holocene is a relatively stable period of climate with temperatures similar to the preindustrial period. Spanning from 11.7 thousand years ago (ka) to the present, the Holocene presents an opportunity to study natural climate variations over thousands of years, illuminating climate variability over timescales much longer than the relatively short instrumental record. The Holocene has been relatively well-sampled by
proxy records, resulting in extensive collections of global data amassing records of temperature (Kaufman et al., 2020a), stable water isotopes (Konecky et al., 2020), speleothems (Comas-Bru et al., 2020), temperature of the Common Era (PAGES2k Consortium, 2017), and more.

Proxy records provide information specific to certain locations, time periods, temporal resolutions, seasons, and climate variables. This presents a multi-faceted but incomplete perspective on past climate. To gain a larger-scale
perspective–and to help account for biases in individual proxy records–it is desirable to synthesize these data into global, hemispheric, or spatial reconstructions of past climate variations. However, large-scale synthesis remains an ongoing challenge in paleoclimate research. Any method that computes global quantities based on location-specific observations must make assumptions about how observed data relates to unknown regions or unsampled climate quantities.

Addressing this challenge, paleoclimate data assimilation provides an intuitive way of fusing paleoclimate
information from proxy data with climate physics, usually provided by a climate model 'prior' in online (e.g., Goosse et al., 2012) or offline (e.g., Steiger et al., 2014; Hakim et al., 2016) approaches. Networks of paleoclimate archives, called proxy data, provide temporal information across multiple sites, while the climate model helps quantify missing information in space using model dynamics and spatial relationships, making the necessary inferences between known data (i.e., temporal evolution of climate at a given location) and unknown data (missing values in space, infilled using climate model
relationships) (e.g., Hakim et al., 2016; Steiger et al., 2018). Output from climate models is used to quantitatively connect proxy locations to other climate variables throughout the globe. In theory, this means that data assimilation can be used to reconstruct the complete climate system based on available information. However, the method should be most skillful for variables that are both (1) closely related to assimilated proxy data and (2) have broad spatial covariances (Steiger et al., 2017). Therefore, reconstructions are often constrained to a subset of climate variables. The goal of paleoclimate data
assimilation is to transform a set of proxy records into a spatially complete and multivariate perspective on climate through time.

Data assimilation-based reconstructions of past climate have nearly all been confined to the Common Era (e.g., Bhend et al., 2012; Goosse et al., 2012; Steiger et al., 2014; Steiger et al., 2018; Hakim et al., 2016; Tardif et al., 2018; Erb et al., 2020; Neukom et al., 2019a, 2019b). Data assimilation has also been used to infer mean temperature of the last glacial
maximum (LGM) from a global collection of sediment cores (Tierney et al., 2020) and very recently to produce a reconstruction from the LGM to the present (Osman et al., 2021). Here, we use data assimilation to reconstruct Holocene temperature using a new multi-timescale reconstruction methodology that seeks to assimilate each proxy record using timescale appropriate spatial covariance patterns instead of using patterns calculated on a single timescale (c.f., Tierney et





al., 2020, Osman et al., 2021). Compared to Osman et al., 2021, for example, we also incorporate a larger proxy database that contains both oceanic and terrestrial proxy records.

## 2. Methods

In paleoclimatic data assimilation, proxy records provide data about past climate at specific locations, seasons, and for different spans of time. Relationships from climate models are used to connect the proxy data to the rest of the globe within a physically-consistent framework. This process requires: (1) a proxy database complete with relevant metadata, (2)
climate model output that realistically quantifies climate relationships for the timeframes of interest, (3) proxy calibrations or proxy system models (PSMs) that relate proxy quantities (e.g., $\delta^{18}$O, tree-ring width) to climate quantities (e.g., temperature), and (4) the update equations of data assimilation that propagate pointwise information spatially to the rest of the climate system (e.g., Steiger et al., 2014, Hakim et al., 2016). These four components are described below.

### 2.1. The proxy database

The Temperature 12k proxy database consists of 1319 temperature-sensitive proxy records from 679 locations across the world (Kaufman et al., 2020a). Each record consists of a time series of data from a specific location along with relevant metadata. We use a slightly updated version of the database (v1.0.2), which contains 713 lake sediment, 359 marine sediment, 193 peat, 26 glacial ice, thirteen speleothem, ten midden, three wood, and two ground ice records. To ensure that proxy records provide sufficient Holocene climatic information, each record generally covers at least 4000 years and meets
age control standards and other criteria (Kaufman et al., 2020a). The dataset contains metadata about location, inferred seasonality, uncertainty, and several other variables; the units of more than 95% of the records are already calibrated to temperature in degrees Celsius. The Temperature 12k dataset has previously been used to compute index reconstructions of mean temperature anomalies for latitude bands and the global mean (Kaufman et al., 2020b).

The Temperature 12k database presents a spatially diverse and multifaceted perspective on Holocene temperature,
but also possesses limitations. It has uneven spatial coverage with large areas of the Southern Hemisphere under-sampled compared to the Northern Hemisphere. Many of the records are seasonally biased, with 34% interpreted to record summer temperature and 20% interpreted to record winter temperature (Kaufman et al., 2020a). Additionally, the Temperature 12k data have uncertainties in the magnitude, timing, and interpretation of the records, which is typical for paleoclimate data. Temperature magnitude uncertainty in the database is quantified for each record as root mean square error (RMSE) based on
archive type, measured proxy, and seasonality (Kaufman et al., 2020a), ranging from 1.1 to 3°C. These values, after being converted to mean square error, are incorporated into the data assimilation so that records with larger uncertainties are given less weight in the reconstruction. For this work, we exclude records that are not calibrated to temperature, lack uncertainty estimates, or do not overlap with the chosen 3-5 ka reference period. Additionally, we do not include seasonal records when an annual record is available for the same location, as in past work (Kaufman et al., 2020b). Of the 1319 proxy records in the





Temperature 12k database, 1276 are calibrated to temperature and 711 are used in the data assimilation. The majority of the
excluded records are seasonal records at locations that also have annual records.

Regarding temporal resolutions, mean resolutions of calibrated proxy records in the Temperature 12k database
range from 1 to over 3000 years, with 96% of the records having mean resolution finer than 500 years. Since the database
lacks precise information about the duration of each data point, we assume that each proxy record contains contiguous data,
with each data point representing an average of the period between data midpoints. This assumption represents one
endmember within a range of possibilities and, because not all proxy datasets are sampled contiguously, this assumption
effectively transfers some higher-frequency variability to lower-frequencies (though we expect this effect to be small).

## 2.2. Climate model data

Paleoclimatic data assimilation fuses proxy data with information from climate models and requires a collection of
climate states drawn from a single simulation or multiple simulations. This ensemble of model states provides two pieces of
information. First, it provides an initial range of climate anomalies for the period of interest, which is later updated through
comparison with proxy data. Second, the model ensemble is used to compute covariances between different locations,
seasons, and climate variables. These covariances allow the method to infer remote climate anomalies based on the location-
specific climate data from proxy data. Because these model data represent our knowledge of the climate system before
assimilating proxy data, it is known as the model "prior."

A good model prior should be relevant to the period of interest, accurately capture realistic relationships in the
climate system, and be long enough to quantify these relationships on paleoclimate-relevant timescales. For Holocene
climate, we draw prior climate states from two transient Holocene model simulations. The first simulation is a PMIP4
HadCM3 transient climate simulation of the past 23 ka. This simulation (also used by Snoll et al., 2022) follows the PMIP4
protocol for the last deglaciation, version 1 (Ivanovic et al., 2016), using the ICE-6G_C VM5a ice sheet reconstruction
(Peltier et al., 2015) and the BRIDGE version of HadCM3 (Valdes et al., 2017), specifically HadCM3B-M2.1dD. The land-
sea mask, bathymetry, and ice-mask are updated at 500-year intervals for the period studied here, in accordance with the
temporal resolution of ICE-6G_C. Orographic changes are applied by linearly interpolating at annual resolution between the
ICE-6G_C time steps. This provides smooth evolution of surface orography and thus reduces the propensity for sudden
climate shocks that can occur if only making these changes at 500-year (or less frequent) intervals, especially at times of
rapid deglaciation (Gregoire et al., 2012, 2016). As recommended by the PMIP4 protocol, freshwater forcing from melting
ice was computed from a high resolution (30-arc seconds) network drainage model of ICE-6G_C (e.g., Wickert et al., 2016)
following the method employed by Ivanovic et al (2017, 2018). Orbital forcing and radiatively active gases ($CO_2$, $N_2O$, $CH_4$)
evolve smoothly, interpolating at annual resolution between any lower resolution time steps of the PMIP4 last deglaciation
forcing dataset. This model simulation is from the latest generation of transient simulations spanning the period. The direct
climate output has undergone light spatial smoothing to account for a minor checkerboard pattern that developed. The





second model simulation used in the prior is the TraCE-21ka simulation, which is an earlier transient simulation spanning from the last glacial maximum to present day and has been described in past work (Liu et al., 2009).

In the data assimilation, output from both the HadCM3 and TraCE-21ka simulations are averaged to decadal resolution and used together in a multi-model ensemble prior. The latitude by longitude resolution for the models is 2.5° by 3.75° for HadCM3 and ~3.71° by 3.75° for TraCE-21ka, but both have been regridded to 2.8125° by 3.75° so they can be used together. The mean of the 3-5 ka period was removed from each model. We composed the prior as all decades within a 5010-year window that was centered, to the degree possible, on the decade to be reconstructed, resulting in a shifting collection of 1002 decades (i.e., 501 decades from two models). The 5010-year length of this window is arbitrary, but was

chosen to be long enough to encompass numerous model states and short enough to allow distinct changes as orbital forcing and boundary conditions evolved through the Holocene. The use of decadal resolution speeds up the data assimilation (compared to annual resolution) and is already equal or higher than the mean resolution of the vast majority (99%) of datasets in the Temperature 12k database. Maintaining decadal resolution also allows for high resolution proxy data to inform the data assimilation on much shorter timescales than previous Holocene data assimilation efforts. Of the 1319

records, only 12 have mean resolution that is sub-decadal, although 236 records have minimum resolutions finer than decadal.

Because the prior climate states are taken from a moving window, both the mean climate and the spatial and seasonal covariance patterns change through the Holocene. Slowly evolving covariance patterns are realistic, so it is appropriate to account for this in the prior. For example, orbital forcing alters seasonal and latitudinal insolation patterns

throughout the Holocene. Additionally, the melting of remnant ice sheets alters spatial climate patterns, particularly on and near the ice sheets. The use of temperature states taken from a moving window allows time-varying relationships to be represented in the prior.

The choice of a time-varying (e.g., Osman et al., 2021) or time-constant prior (e.g., Hakim et al., 2016) is an important consideration in offline data assimilation. Whether the prior varies in time or not, the temporal evolution of the

model prior will influence the reconstructed climate, so it should be chosen carefully. A time-varying prior, as used here, may impart some aspects of its temporal evolution onto the final reconstructed climate. To test how the reconstructed climate is affected by the model prior and other methodological choices, alternate experimental designs are explored in Appendix B.

### 2.3. Proxy calibrations

In data assimilation, proxy records must be quantitatively compared to model values in the same units. This can be

done using empirical methods such as linear regression (e.g., Hakim et al., 2016), physically based proxy-system models (e.g., Dee et al., 2016, Tierney et al., 2020), or other approaches. In the work presented here, most of the Temperature 12k proxy records have already been calibrated to temperature (Kaufman et al., 2020a), so we rely on those previous proxy calibrations rather than a proxy-system modeling approach, primarily because of the lack of robust PSMs for most of the data in the Temperature 12k compilation.





**2.4. Multi-timescale data assimilation**

Data assimilation is a mathematical technique for optimally combining observations (here proxy data) with prior information, typically from a model. The model is a climate model that provides an initial, or prior, state estimate that can be updated in a Bayesian sense based on the information from the proxies and error estimates of both the proxies and the prior. The prior may contain any climate model variables of interest and the updated prior, called the posterior, is a probabilistic estimate of the true climate state given the observations and the error estimates. The basic data assimilation state update equations (e.g., Kalnay 2003) are given by

$$x_a = x_b + K[y - H(x_b)] \tag{1}$$

where **K** is the Kalman gain matrix, which can be written as

$$K = cov\big(x_b, H(x_b)\big)\big[cov\big(H(x_b), H(x_b)\big) + R\big]^{-1} \tag{2}$$

and cov represents a covariance expectation. The matrix $\mathbf{x}_b$ is the prior (or "background") estimate of the state and the matrix $\mathbf{x}_a$ is the posterior (or "analysis") state and represents the ensemble reconstruction. Observations or proxies are contained in the vector **y** and the observations are estimated by the prior through $\boldsymbol{H}(\mathbf{x}_b)$, which is an operator that maps $\mathbf{x}_b$ from the state space to the observation space (e.g., converts climate model variables to measured proxy quantities). Note that here we assimilate only proxies that have already been converted to units of degrees Celsius, so our $\boldsymbol{H}(\mathbf{x}_b)$ is simply an ensemble of temperature values from $\mathbf{x}_b$ at the same locations, seasons, and temporal resolutions as the proxy records. The difference between **y** and $\boldsymbol{H}(\mathbf{x}_b)$ represents the new information added by the proxies. From the first term of Eq. 2, we see that **K** is fundamentally a spatial covariance matrix that spreads the information added by the proxies, $\mathbf{y} - \boldsymbol{H}(\mathbf{x}_b)$, to all variables in the prior $\mathbf{x}_b$. **R** is a positive and diagonal error covariance matrix for the proxies, where each diagonal element is the error term for each proxy. As the values of **R** become large, corresponding to higher proxy uncertainties, **R** comes to dominate the matrix inverse in Eq. 2 which, because it is positive and diagonal, leads to a **K** that approaches zero; thus, in a high proxy-uncertainty scenario, $\mathbf{x}_b$ is modified only slightly. In a low proxy-uncertainty scenario, the opposite situation occurs, the new proxy information is weighted more heavily, and $\mathbf{x}_b$ is modified more substantially. The data assimilation process involves computing the above equations, which "updates" the prior $\mathbf{x}_b$ to arrive at the posterior state $\mathbf{x}_a$ for each timestep. For paleoclimate data assimilation, the reconstruction consists of $\mathbf{x}_a$ computed for each fundamental time step of the reconstruction (e.g., every decade of the Holocene). As in Steiger et al., 2018, Eqs. 1-2 are implemented using a square root ensemble Kalman filter outlined in (Whitaker & Hamill, 2002). See Steiger et al., 2014 and Steiger and Hakim, 2016 for detailed interpretations of the data assimilation update equations.

Here, we expand from the multi-timescale data assimilation approach developed by Steiger and Hakim (2016). Multi-timescale data assimilation is distinguished from single timescale data assimilation (used by all previous data assimilation-based paleoclimate reconstructions) in that multiscale proxy data are assimilated using timescale appropriate covariances rather than covariances calculated at a single uniform resolution (e.g., Badgeley et al., 2020; Osman et al., 2021). Such a multiscale approach allows us to utilize covariances across timescales to update the reconstruction (Steiger





and Hakim 2016). This is important in a scenario where, for example, high-frequency and low-frequency covariances between locations differ or even oppose each other. Also, a multiscale reconstruction approach can reduce the chances of

obscuring climate signals in the proxy data because it does not impose a single "sampling" time scale on all proxy data regardless of their true time resolution.

Here, we update the methodology of Steiger and Hakim 2016 by modifying two components: (1) We have a different technique for creating and structuring the multi-timescale prior, $\mathbf{x}_b$, as well as $H(\mathbf{x}_b)$; (2) We additionally employ a simultaneous square root Kalman filter (all observations at a given time step are assimilated simultaneously) instead of a

sequential square root Kalman filter. These primarily technical modifications result in an algorithm that is faster and requires far less memory storage (a major limiting factor in the Steiger and Hakim 2016 algorithm) to complete the reconstruction. We note that, given the same inputs, simultaneous and sequential assimilation techniques produce identical ensemble means and only minor differences in ensemble spread.

In this multi-timescale data assimilation approach, the reconstructions are performed off-line at a predetermined

base timescale, here decadal resolution, though the algorithm is general and can apply to any base timescale (e.g., annual or centennial). To process all proxy data to decadal resolution, we first average any sub-annual data, then generate values for every year using nearest-neighbor interpolation, and finally bin these annual values to decadal means. As stated earlier, this processing makes the assumption that proxy data is continuous (unless NaNs are present). A uniform temporal resolution is necessary to perform the data assimilation, but information about the time-resolution of each proxy data point is retained and

proxy data is assimilated using timescale appropriate covariances.

$\mathbf{x}_b$ is composed of base time scale averaged climate states, taken from a climatically representative climate model simulation (or simulations, described previously). $H(\mathbf{x}_b)$ is pre-computed for each proxy over the full set of temporal resolutions contained within that proxy time series; the time averages for $H(\mathbf{x}_b)$ are computed such that the center year of $\mathbf{x}_b$ and $H(\mathbf{x}_b)$ are the same (or half a time step away for a span of an even number of time steps). For example, suppose that $\mathbf{x}_b$ is

composed of 10-year averaged climate states and a proxy has values that are decadal means or multi-decadal means of different lengths; we pre-compute multiple ensembles of $H(\mathbf{x}_b)$ for this proxy: one ensemble uses 10 year-averages of climate model data and the others use multi-decadal means computed with box averages of decades centered to the degree possible on the same decades. Relationships between the ensembles of $H(\mathbf{x}_b)$ and $\mathbf{x_b}$ quantify how proxy estimates (at the same location, season, and timescale as the proxy data) relate to decadal-mean climate everywhere else. Note that in

methodological tests we tried different choices of averaging windows, associating each $H(\mathbf{x}_b)$ with only time steps prior to or after the center time step of $\mathbf{x}_b$, but we found that center-referencing performed best.

The data assimilation update equations are then computed for each base time step in turn, assimilating all proxies that have values spanning a given time step. For proxies with a resolution lower than the base timescale, the proxy value will be assimilated repeatedly for all the base time steps it spans (e.g., a value spanning the years 1000 to 1050 BP will be

assimilated at each decadal time step within that time range); thus, this repeated assimilation updates the entire period that a proxy value represents in base time step segments. The reconstruction code uses the pre-computed $H(\mathbf{x}_b)$ which applies for





the particular time average of a given proxy value. Depending on the time resolution of a proxy data point, $\boldsymbol{H}(x_b)$ values can represent time averages ranging from 10 years (our chosen base resolution) to 1000 years. Because of our use of center referencing and temporal averages of up to 1000 years, no ensemble members will be centered on decades more recent than

500 years before present in the simulations. The temporal resolution is capped at 1000 years to prevent the loss of additional ensemble members at the modern end of the simulation.



Figure 1. Temperature anomalies from calibrated records. Relative temperatures (°C) for 1263 calibrated proxy records in the Temperature 12k database. Records are interpolated to decadal resolution using nearest-neighbor interpolation and arranged

from north to south. The 3-5 ka mean is removed from each record, and the thirteen records that do not have data between 3-5 ka show no data. Black lines indicate the timing of the warmest decade for each 30° latitude band, calculated by standardizing all records within each latitude band and finding the warmest mean where at least 25% of the proxy records have values. The y-axes show how many records are displayed (left) and the approximate latitudes (right).




## 3. Results

### 3.1. Proxy network analysis


The Temperature 12k dataset contains 1319 proxy time series and substantial metadata and has been described and synthesized into global means in past work (Kaufman et al., 2020a; Kaufman et al., 2020b). 1276 of these proxy records have been calibrated to degrees Celsius, and 711 are used in the data assimilation-based reconstruction. To visualize this data-rich network, calibrated proxy records are plotted from north to south with each proxy represented as a color-coded line

(Fig. 1). This perspective allows the entire database to be visualized at a glance, although it ignores fundamental aspects of the data such as longitude and seasonality.

The calibrated proxy records show considerable spatial and temporal variability, but some consistent patterns emerge. The early Holocene is cold in most records, with many records showing warming toward the mid-Holocene. Maximum preindustrial temperatures typically occur around 6-7 ka in the Northern Hemisphere, with the late Holocene

showing cooler temperatures. There is also considerable spatial and temporal variability, some of which may represent genuine temperature variability while some may represent noise. To quantify temperature trends in the data, we perform a linear regression of the original proxy data, using a Wald Test and a t-distribution with $p \leq 0.05$ to determine significance (Table 1). Using this metric, more proxy records show significant warming as opposed to cooling from 12 ka to 6 ka (49.6% warming vs. 14.4% cooling), while more records show significant cooling than warming from 6 ka to 0 ka (40.9% cooling

vs. 15.3% warming).

| | 12 - 6 ka | | | 6 - 0 ka | | |
|---|---|---|---|---|---|---|
| | *Warming* | *Cooling* | *Flat* | *Warming* | *Cooling* | *Flat* |
| All | 49.6 | 14.4 | 36.0 | 15.3 | 40.9 | 43.7 |
| Annual | 53.7 | 12.2 | 34.2 | 15.9 | 38.5 | 45.5 |
| Summer | 45.3 | 16.2 | 38.5 | 11.7 | 48.2 | 40.1 |
| Winter | 46.9 | 16.6 | 36.5 | 19.9 | 34.5 | 45.6 |

**Table 1. Percentage of records with given temperature trends. Linear regression slopes are calculated for the periods 12-6 ka and**
**6-0 ka and the percentage of records which fall into each category are listed. "Flat" refers to records that fail the Wald Test for slopes significantly different from 0 at a 0.05 level, using a t-distribution. The 1276 calibrated records are considered, with 587 annual records, 427 summer records, and 262 winter records. Records with fewer than 5 points in a given period are excluded from that period, which accounts for no more than 20% of records in each category. Percentages may not sum to 100 due to rounding.**






The dataset possesses both spatial and seasonal biases, however, so summary statistics should not be taken as straightforward indicators of global mean temperature trends. An important consideration when examining Holocene temperatures is the possible effect of seasonal biases in proxy data, which is especially important considering time-varying external climate forcings over the Holocene. Changes in aspects of Earth's orbit, for instance, redistribute incoming solar

radiation between seasons and latitudes, producing different trends in insolation for different seasons and locations. From the early Holocene to present day, Northern Hemisphere insolation has decreased in the summer, increased in the winter, and remained relatively stable for the annual mean (Fig. 2). Climate feedbacks, which can amplify or diminish the climate response to climate forcings (Erb et al., 2013), may further modify seasonal signals, so care must be taken not to misinterpret a seasonal proxy signal as an annual mean. When using metadata to separate records by season, many proxy records of all

seasons show warming in the early Holocene and cooling in the late Holocene, although most of these records are in the Northern Hemisphere (Fig. 3). Interestingly, summer records have the highest percentage of time series with clear late Holocene cooling (48.2%), while annual and winter records have a plurality of time series without significant trends (only 38.5% of annual and 34.5% of winter records show cooling; Table 1). This is somewhat consistent with insolation forcing. In data assimilation, the use of a time-varying prior helps account for changing relationships between seasonal proxy signals.

Additionally, sensitivity tests conducted in Sect. 3.3 can be used to evaluate the extent to which our seasonal interpretation of proxy records can affect the final reconstruction.

In addition to showing general temperature patterns, these overview figures illustrate the predominance of Northern Hemisphere records (over half of the time series are in the Northern Hemisphere mid-latitudes) and the truncation of many records near 11 ka, a result of the processing conducted for a previous pollen proxy synthesis effort (Marsicek et al., 2018).

Many other aspects of the database, such as proxy type, longitude, season, uncertainty, and other metadata, are not shown. Additional analysis of the proxy database can be found in recent publications (Kaufman et al., 2020a; Kaufman et al., 2020b) and the proxy data can be visualized online at lipdverse.org.

## 3.2. The past 12,000 years

We now assimilate the Temperature 12k proxy database. The model prior uses decadal climate anomalies from two

models, the transient HadCM3 and TraCE-21ka simulations. Because the prior uses climate states from a moving 5010-year window, the mean and the covariance patterns change through time (Fig. 4). Starting from this prior, we assimilate the Temperature 12k proxy records to produce a spatially complete reconstruction of Holocene temperature from 12 ka to the present.




**Figure 2. Modeled hemispheric insolation and temperature in different seasons. Insolation (W m⁻², dashed) and temperature (°C, solid) from the HadCM3 deglacial simulation, averaged for the annual mean (black), June-August (red), and December-February (blue) for the (a) Northern Hemisphere and (b) Southern Hemisphere. Throughout this paper, original monthly data is used from models, with no adjustment to account for the calendar effect (Joussaume and Braconnot, 1997).**






**Figure 3. Temperature anomalies for calibrated proxy records separated by seasonal metadata. (a-c) Maps of proxy record locations, separated by season. (d-f) Relative temperatures of calibrated records, as in Fig. 1, separated by season.**





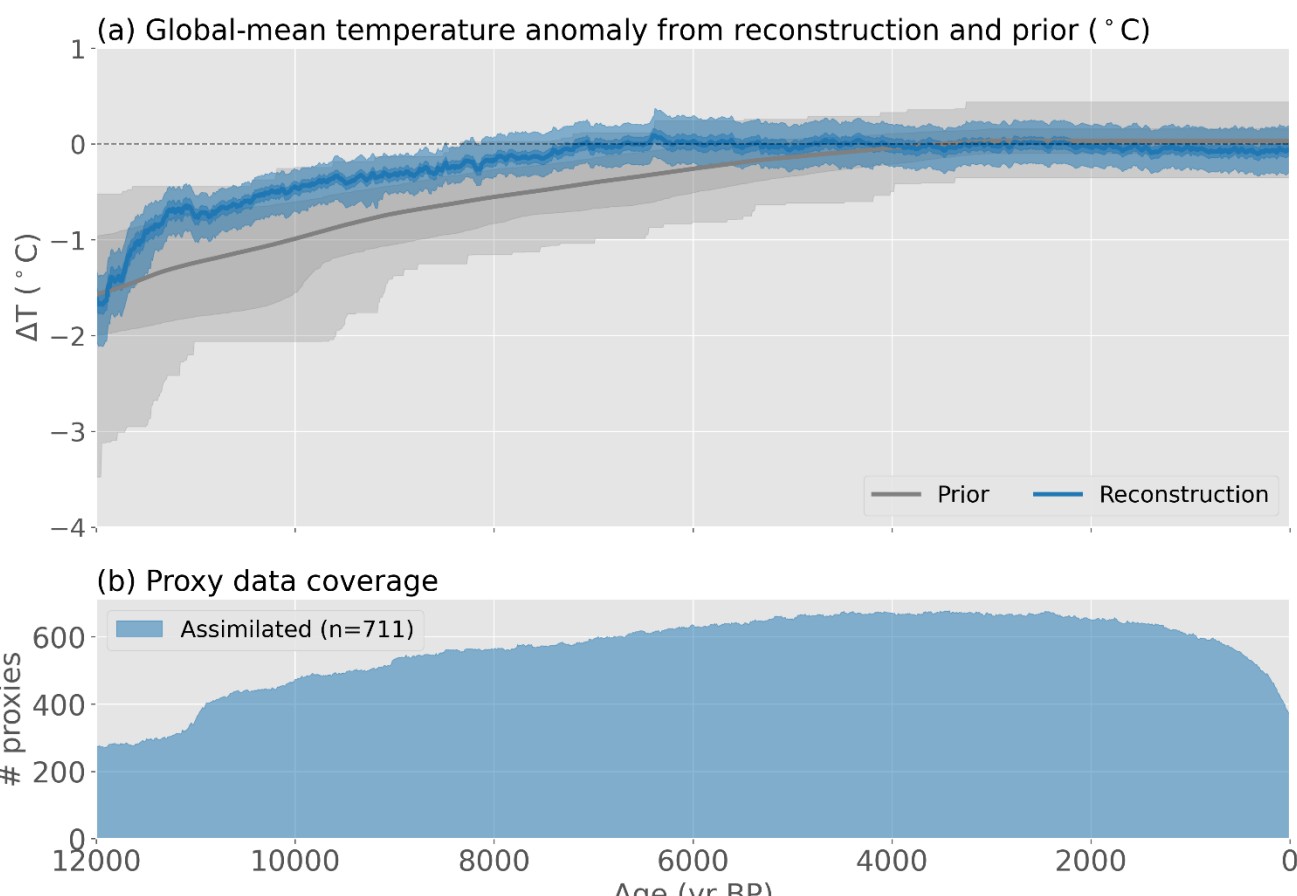

**Figure 4. Reconstructed global mean temperature. (a) Global mean temperature in the prior (gray) and reconstruction (blue). Lines show the ensemble mean and colored bands show the 1-sigma and full ranges of the ensemble. The reconstruction uses 3-5 ka as the reference period, as most records overlap with that period. (b) The temporal proxy coverage, showing the number of assimilated observations at each time step.**

Reconstructed global mean temperature warms rapidly at the end of the last glacial period, with ~1.2°C warming from 12 to 10 ka (Fig. 4). The temperature shift near 11 ka is likely a result of the rapid increase in proxy coverage at that moment. After 10 ka, warming continues at a slower pace with peak warmth around 6.4 ka followed by a gradual cooling toward present day. Note that some reconstructed values in the early to mid-Holocene are warmer than the climate states in the model prior, demonstrating that reconstructed values can exceed the limits of the prior if proxy values support such anomalies. Proxy coverage is highest near 3-4 ka (~700 records) and lowest before 11 ka (~300 records). The relatively fast 20th century warming seen in the instrumental temperature record is not captured by the reconstruction due to the coarse temporal resolution of the assimilated records (having a mean resolution of ~200 years) and the decrease in proxy coverage toward the present day.





320   Spatially, the reconstruction shows warming in the first half of the Holocene over almost the entire globe, with some of the largest values over the regions of disappearing ice sheets in the Northern Hemisphere (Fig. 5a). Changes are generally larger over continents than over the ocean and tend to be larger over the Northern Hemisphere than the Southern Hemisphere, although in pseudoproxy tests, the skill of the reconstructed temperature is reduced in the Southern Hemisphere relative to the Northern Hemisphere (Appendix A). For the latter half of the Holocene, temperatures decrease in most of the

325 Northern Hemisphere, with notable exceptions in regions of India and northern Africa where stronger mid-Holocene monsoons may have allowed for cooler mid-Holocene climate (Brierley et al,. 2020; Fig. 5b). Southern Hemisphere temperature changes are small in the late Holocene.

   To better understand the temporal and spatial characteristics of the Holocene reconstruction—and to further explore the complexity of the underlying proxy network—we here compare the reconstructed climate to the proxy records that

330 inform it. This proxy/reconstruction comparison helps illustrate how the multifaceted proxy data is transformed into a spatially complete product.

   Temperature trends are first compared for the reconstruction and individual proxies in the data-rich regions of North America and Europe (Fig. 6). Notably, the reconstructed temperature anomalies are more spatially uniform than those seen in the proxy records themselves. Proxy records are diverse and sometimes contradictory, with temperature trends that vary

335 substantially even over short distances. These spatially diverse climate signals are impossible to fully match using the relatively coarse spatial resolution of the data assimilation (2.8125° latitude by 3.75° longitude). Additionally, the data assimilation is constrained by the model's spatial covariance pattern, which prohibits unrealistically large changes over short distances. Consequently, the data assimilation product often serves as an effective compromise between opposing and high amplitude anomalies in a region. A side effect of this compromise is that the reconstructed temperature often cannot match

340 the large positive and negative anomalies of proxies in the region. Resolving the cause of this apparent spatial variability in the proxy database—whether it represents real spatial differences, proxy interpretation uncertainty, age model uncertainty, or some other source of uncertainty—should continue to be a research priority.

   To compare the temperature reconstruction and proxy records in a different way, the reconstructed zonal mean is compared to annual proxy values binned into half-degree latitude bands (Fig. 7). This helps reveal the extent to which the

345 reconstruction matches—or fails to match—the complex spatial and temporal patterns of the proxy data. The Holocene Reconstruction shows some clear similarities to the annual mean proxy data, with coldest temperatures in the early Holocene and warmest temperatures later. Again, the Holocene Reconstruction is more spatially and temporally homogeneous than the data. The reconstruction shows warmest temperatures close to 6 ka in the Northern Hemisphere mid and high latitudes, where proxy coverage is densest, but more recent in the tropics and Southern Hemisphere. Reconstructed temperature

350 anomalies are largest in the northern mid and high latitudes as well as the Antarctic region, with relatively small temperature anomalies in the tropics and southern mid-latitudes. Part of the reduced Southern Hemisphere signal may be indicative of the real climate, as the Southern Hemisphere has larger oceans and is more remote from changes in Northern Hemisphere ice sheets, although the relative lack of sufficient proxy data in the Southern Hemisphere likely also contributes to this result.






**Figure 5. Temperature trends.** Temperature trends (°C kyr$^{-1}$) at every location for the periods (a) 12 to 6 ka and (b) 6 to 0 ka. Dots show locations of assimilated proxy records during each period. Note the different scales between the two panels.



**Figure 6. Temperature trends in the reconstruction and proxy records for North America and Europe. Temperature trends (°C kyr⁻¹) over the periods (a) 12 to 6 ka and (b) 6 to 0 ka, like Fig. 5. Trends in assimilated proxy records are shown as colored symbols, with shapes indicating the seasonality of the proxy (circle: annual; upward-pointing triangle: summer; downward-pointing triangle: winter). Records are only plotted if they have data covering at least half of the time period. Note the different color bars in the two panels.**






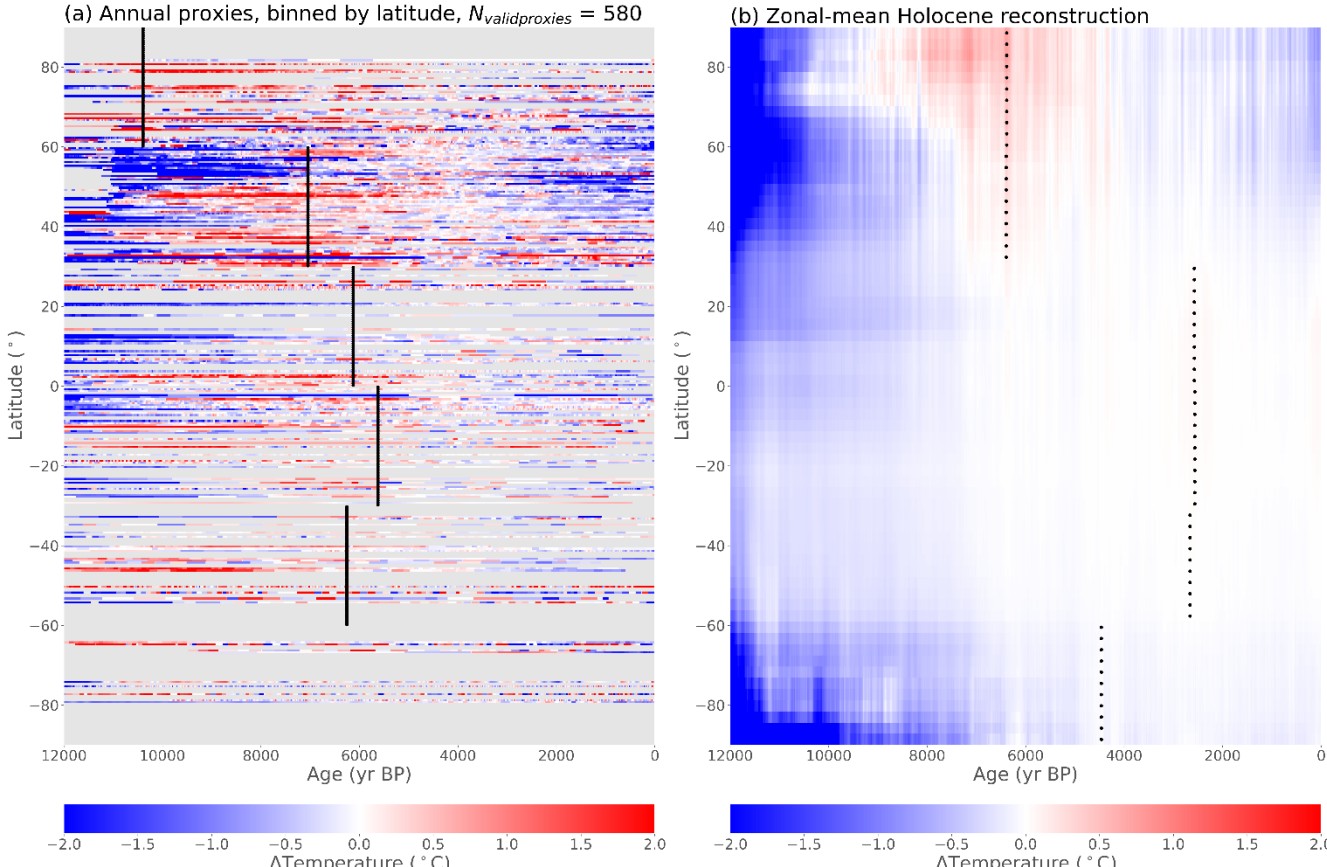

**Figure 7. Comparison of annual records and zonal mean Holocene Reconstruction. (a) Annual Temperature 12k proxy records binned into 0.5° latitude bands, showing temperatures relative to 3-5 ka. (b) Zonal mean annual temperatures in the Holocene Reconstruction. Panel (a) is like Fig. 1, but only annual records are selected and they are binned by latitude and averaged when multiple records occupy the same latitude band. Panel (a) does not represent zonal means, whereas (b) shows the zonal mean of the reconstruction. Black lines or dots show the timing of the warmest period, calculated by standardizing all latitude bands and finding the warmest mean of each 30° latitude zone where at least 25% of the bands have values.**

### 3.3. Possible influence of proxy seasonal biases

Changes in Earth's orbit affect Holocene insolation trends differently in different seasons (Fig. 2). Since the early to mid-Holocene, insolation in the Northern Hemisphere has decreased in summer, increased in winter, and stayed relatively stable in the annual mean. These seasonal insolation trends affect seasonal temperature, with warmer early to mid-Holocene temperatures in Northern Hemisphere summer and colder temperatures in winter relative to the annual mean in the HadCM3 transient simulation (Fig. 2). In the Southern Hemisphere, a similar but opposite insolation pattern occurs, but seasonal temperatures are less impacted due to the large ocean basins.



The existence of differing seasonal temperature trends highlights the need to accurately diagnose seasonal biases in
proxy records. If summer biased proxy records are assumed to represent annual means, for example, reconstructed temperatures may show too much early to mid-Holocene warmth. If winter biased proxies are used instead, the opposite is true.

Data assimilation accounts for proxy seasonality directly by transforming seasonal proxy values into annual means using covariance relationships between seasonal and annual values in the model prior. To do this, the method requires
accurate seasonality metadata for assimilated proxies. If metadata about proxy seasonality is inaccurate, then season-specific temperature trends may still bias the final reconstruction.

In our main reconstruction, we use seasonality metadata from the Temperature 12k database. Assimilated proxies are prescribed to be 78% annual, 21% summer, and 1% winter. To explore the extent to which incorrect seasonality metadata could bias results, we run three additional experiments. In the first experiment, all proxies are assumed to represent summer
values: June-August in the Northern Hemisphere and December-February in the Southern Hemisphere. In the second experiment, all proxies are assumed to represent winter values in their respective hemispheres. In the final experiment, proxy values are assumed to represent annual means. The proxy data are not modified; we only change how assimilated proxy data are translated into annual mean reconstructed temperature.

In the "summer" experiment, reconstructed annual mean temperatures become cooler in the early to mid-Holocene,
with a value of 0.02 °C at the mid-Holocene (compared to 0.09 °C in the default experiment) (Fig. 8). This reduction in early to mid-Holocene warmth is consistent with expectations for the Northern Hemisphere, where summer temperatures were relatively warm in the early to mid-Holocene. When accounting for this possible bias, the reconstructed annual mean temperature in the early to mid-Holocene becomes cooler. In comparison, the "winter" experiment–which assumes that all proxies represent winter values in their respective hemispheres–produces an opposite response: accounting for the relative
cold of the early to mid-Holocene winter produces an annual mean reconstruction that is warmer during that period, with a mid-Holocene anomaly of 0.17 °C, nearly twice that of the default experiment. In both the "summer" and "winter" experiments, changes to global mean temperature trends are more affected by anomalies in the Northern Hemisphere than the Southern Hemisphere because, despite similar but opposite insolation patterns in the Southern Hemisphere, seasonality changes in that region are damped due to large oceans (Fig. 2).

These results show that our perception of Holocene trends can be influenced by assumptions about proxy seasonality. If proxy records have an undiagnosed summer bias, some of the mid-Holocene warmth in climate reconstructions may simply represent summer warmth. On the other hand, if proxy records have an undiagnosed winter bias, mid-Holocene warmth could be even greater than reconstructions show. That said, even in these extreme scenarios where all proxies are assumed to represent either summer or winter anomalies, reconstructed mid-Holocene temperatures only differ
by a couple tenths of a degree, not nearly enough to match the cold mid-Holocene anomalies present in transient climate simulations (-0.56°C in HadCM3 and -0.29°C in TraCE-21ka). Therefore, proxy seasonality biases can potentially explain part of the Holocene temperature "conundrum" (Liu et al., 2014), but by no means all of it.



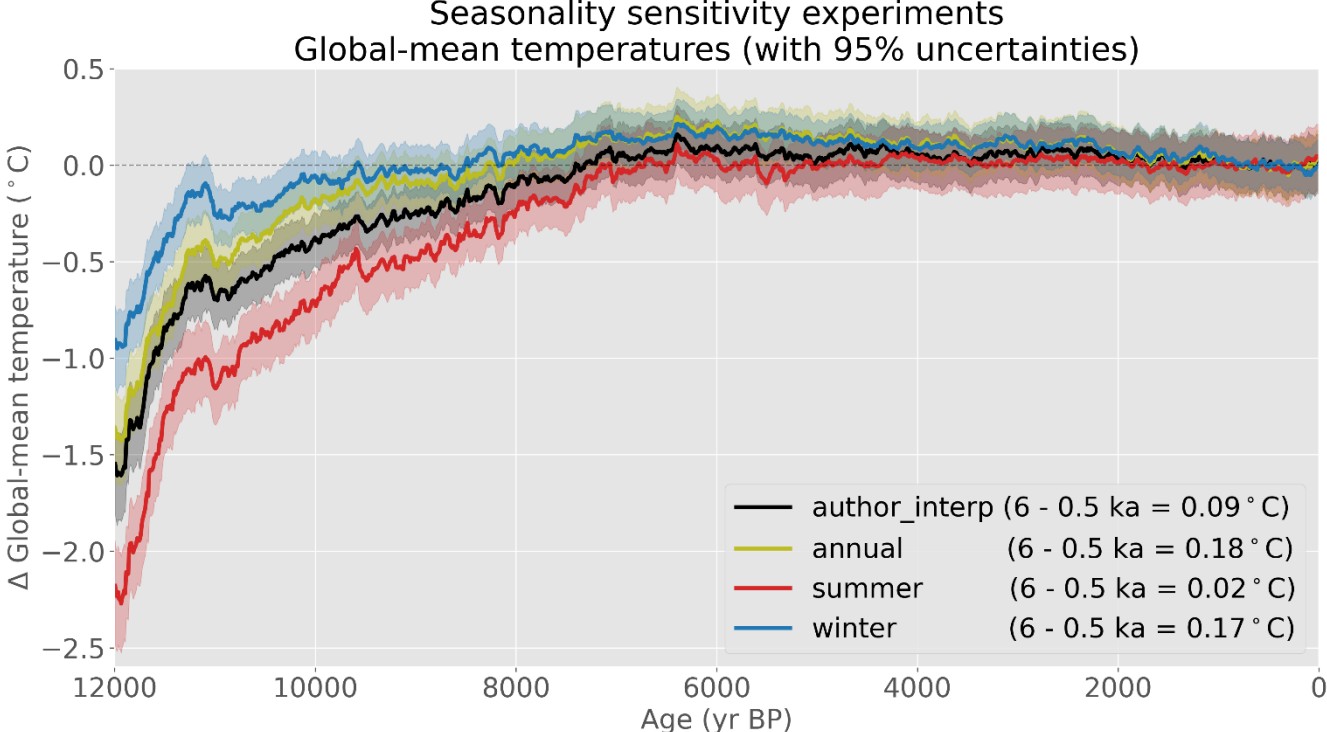

**Figure 8. Global mean temperature composites created using different assumptions for proxy seasonality. "author_interp" is the default experiment, where proxy seasonality metadata from the Temperature 12k proxy database is used. In the other experiments, all proxies are assimilated as if they represent annual, summer, or winter means. Temperature anomalies are shown relative to 0-1 ka.**

## 4. Discussion

### 4.1. Regional comparison with proxy data

To explore the spatial and temporal patterns of the reconstruction in more depth, time-varying temperature anomalies are explored in North America and Europe. These regions are well-covered by proxy records and, since they are locally forced by the shrinking Laurentide and Fennoscandian ice sheets, they present worthwhile targets for closer analysis. Reconstructed North American temperatures are averaged into millennial means spanning the past 12 ka and are plotted alongside ice sheet anomalies and annual mean proxy values binned to the same grid as the reconstruction (Fig. 9). This comparison allows us to examine how the proxy records are translated into the final spatiotemporal temperature reconstruction. Additionally, the ice sheet reconstruction (ICE-6G_C; Peltier et al., 2015) allows us to evaluate the reconstructed temperature patterns against a clear spatial forcing.

As noted previously, the temperature reconstruction shows some agreement with the proxy data but also shows greater spatial uniformity. Widespread cold anomalies exist over North America at 11-12 ka, which reduce in extent and



magnitude as the Laurentide ice sheet shrinks. During 9-11 ka, reconstructed warmth over part of northern Canada is likely caused by the assimilation of warm proxies in nearby western North America but is probably incorrect due to the presence of the Laurentide ice sheet in that region. By 7-8 ka, the effect of the ice sheet appears to be relatively local. By 6-7 ka, cool temperatures have largely disappeared despite some ice remaining in northeast Canada according to the ICE-6G_C ice sheet

reconstruction (Peltier et al., 2015). No proxy records exist for temperatures over extinct ice sheets, so temperatures in those regions are inferred based on available records and covariance patterns from the model prior.

**Figure 9. Temperature and ice sheets in North America. Millennial anomalies for reconstructed temperature (shaded, °C), gridded annual mean proxy records (dots, °C), and ice sheets from the ICE-6G_C reconstruction (contours, 500 m interval; Peltier et al., 2015). All values, including ice sheets, are shown for 1000 year means relative to the period 3000-5000 years before present. Proxy values are binned and averaged to the same spatial resolution as the data assimilation for clarity. Summer and winter-biased proxy records—which make up 21% and 1% of the assimilated records, respectively—are not shown, as seasonal records are not directly comparable to the annual reconstruction. The temperature reconstruction is based on proxy values and model covariances, without knowing the specific timing of ice sheet changes.**





**Figure 10. Temperature and ice sheets in Europe. Millennial anomalies for reconstructed temperature (shaded, °C), gridded annual mean proxy records (dots, °C), and ice sheets from the ICE-6G_C reconstruction (contours, 500 m interval), as in Fig. 9.**





In Europe, cool temperatures prevail until around 8 ka, past the end of the Fennoscandian ice sheet (Fig. 10).

Afterward, temperatures over Scandinavia reach a peak from 5-7 ka before gradually cooling toward pre-industrial temperatures. Reconstructed temperatures in the Greenland Sea show pronounced warmth during 9-11 ka and afterward, which appears to be informed by several records on Svalbard and the waters west and south of Svalbard. The two sediment core foraminifera records from the Fram Strait west of Svalbard (MSM5/5-723-2 and MSM5/5-712-2; Fig. 4f-g of Werner et al., 2016) reflect subsurface (100m depth) temperatures and are likely influenced by increased Atlantic Water advection as

well as the summer insolation peak and limited sea-ice extent across this region during the Early Holocene (Werner et al., 2016?). If they do not correspond with surface temperatures, it may be beneficial to remove these records (and similar ones) from future data assimilation.

In both the North American and European regions, proxy data shows a greater diversity of signals compared to the larger-scale patterns of the reconstruction. Data assimilation represents a best-fit solution given the model, the data, and their

uncertainties.

## 4.2. Northern Hemisphere cooling at 8.2 ka

Evidence from proxy records indicates the existence of a brief cold event near the North Atlantic region around 8200 years ago (Alley et al., 1997, Thomas et al., 2007, Morrill et al., 2013), possibly caused by freshwater influx in the North Atlantic. This event, which has also been studied in models (Tindall & Valdes, 2011, Morrill et al., 2014, Matero et

al., 2017), represents a pronounced multi-decadal climate event that is (at least partially) captured in our Holocene reconstruction. Because of its short timescale and relative age, it is a worthwhile target for further exploration.

In our reconstruction, global mean temperature shows a brief cold excursion for ~100 years near 8.2 ka (Fig. 11). Spatially, the coldest temperatures in the reconstruction occur above the Laurentide ice sheet, with moderate cooling over the Northern Hemisphere mid and high latitudes and mild warmth in parts of the Southern Hemisphere, particularly near

Antarctica. This temperature pattern is generally consistent with data syntheses and climate model experiments of the 8.2 ka event (Morrill et al., 2013), which suggests that the multi-timescale assimilation technique can reasonably reconstruct short-term phenomena, even when only a small fraction (24%) of the assimilated records have the resolution to meaningfully contribute. Although the pattern is generally consistent with previous reconstructions and simulations, there are some key differences. Generally, maximum 8.2 ka event cooling is thought to have occurred in the North Atlantic (Morrill et al.,

2013), in part due to the hypothesis that the event is driven by freshwater forcing in the region (e.g., Matero et al., 2017). Our reconstruction does show substantial cooling in the North Atlantic, but the maximum cooling occurs further west near the remnants of the Laurentide Ice Sheet. This is likely due to our methodology, which uses a prior drawn from a moving 5010 year long window centered on each decade of this event—a period of large changes in the remnant Laurentide ice sheet over the present-day Hudson Bay. Additionally, the method has no information about the exact timing of freshwater forcing

events. For data assimilation to better capture the spatial details of the 8.2 ka event, it may need more specific information about the climate forcing; however, doing so may bias the result to the expected response, which is also problematic.







**Figure 11. Cooling at 8.2 ka. (a) Reconstructed temperature anomalies (°C) for (a) the global mean, (b) spatial patterns for the 8.2 ka event in the reconstruction (shading) and proxies (symbols), and (c) global and regional means at the 8.2 ka event calculated**
**across ensemble members. The periods used for the calculations used in (b,c) are shown in (a), with the anomaly period shown in blue and the reference periods shown in red. Proxies are only shown in (b) if they have at least one value in each of the three periods shown in (a), which is 169 of the 711 assimilated proxies. Proxy seasonality is annual (circles), summer (upward-pointing triangles), or winter (downward-pointing triangles). The spatial extent of the Greenland and Europe regions in (c) are shown in (b). In (c), all ensemble members are shown for the global mean while a randomly selected group of 100 ensemble members are**
**shown for the other two regions.**





Although the temporal pattern is similar, the amplitude of the cooling reconstructed by data assimilation is less than previous estimates. For example, cooling in the Greenland and European regions (-0.47°C and -0.12°C, respectively; Fig. 11c) is less than those seen in proxy-only studies (e.g., -2.2°C and -1.1/-1.2°C, respectively, in Morrill et al., 2013). This is

an expected result, as (in comparison to Morrill et al., (2013)) no effort was made to align the event across age-uncertain records. Age uncertainties in proxy records are often larger than the duration of short events, and assimilation of temporally displaced records will mask or diminish the true extent of the event. This difficulty has been addressed in other studies through the alignment of age models (e.g., Thomas et al., 2007) or by searching for climate excursions within a larger multi-century window (e.g., Morrill et al., 2013), but this still poses a problem for data assimilation, which has so far not been used

with proxy age alignment.

Ultimately, the 8.2 ka event provides a useful test case for exploring the utility and limitations of paleoclimate data assimilation, and provides food for thought for future studies. Adjustments in the model prior, the age models of proxy data, or the temporal resolution of the reconstruction (e.g., Osman et al., 2021) may help account for these issues, but the exact design of these solutions is left to future work.

**4.3. Comparison with past reconstructions**

Previous reconstructions of Holocene temperature have employed an assortment of reconstruction techniques, with many showing peak warmth in the early to mid-Holocene and a clear cooling toward present day (Kaufman et al., 2020b; Marcott et al., 2013; Shakun et al., 2012). This contrasts with transient model simulations, which show warming throughout the Holocene (Liu et al., 2014). Two exceptions to this pattern were published recently. The first, which reconstructs sea

surface temperatures between 40°S and 40°N, attempts to remove a possible seasonal bias by examining proxy trends during the last interglacial (Bova et al., 2021), resulting in a 40°S-40°N sea surface temperature reconstruction which warms throughout the Holocene. The other study uses data assimilation based on marine sediments to reconstruct spatial temperature anomalies since the Last Glacial Maximum, also resulting in warming through the Holocene (Osman et al., 2021).

The mid-Holocene temperature anomaly in those reconstructions, calculated as the difference between the millennia centered on 6 and 0.5 ka, is 0.54°C for Marcott, 0.44°C for Kaufman, -0.27°C for Bova, and -0.17°C for Osman. For comparison, the Holocene Reconstruction presented in this paper shows mild mid-Holocene warmth (Fig. 12) with a temperature anomaly of 0.09°C, fitting between these previous reconstructions but with the central estimate still showing a warmer mid-Holocene. This mid-Holocene warmth, despite being mild, is notable because it emerges when using a time-

varying prior with a colder mid-Holocene. In other words, the initial baseline climate state (from the models) has a colder mid-Holocene, but the proxy data is strong enough to reveal mid-Holocene warmth in the final reconstruction (Fig. 4).



**Figure 12. Comparison of Holocene temperature reconstructions. The Holocene temperature reconstruction using data assimilation (black; this study) compared to other proxy or DA-based reconstructions: Shakun (dark blue; Shakun et al., 2012), Marcott (light blue; Marcott et al., 2013), Kaufman (green; Kaufman et al., 2020b), Bova (olive; Bova et al., 2021), and Osman (red; Osman et al., 2021). All curves represent global means except for the Bova curve, which represents sea surface temperatures between 40°S and 40°N (these quantities are not directly comparable, but are plotted together for convenience). The mean or median (lines) and 1 sigma uncertainty values (shaded) are shown for all reconstructions, and the 95% range is also shown for the new Holocene Reconstruction. The Temperature 12k reconstructions consist of five different reconstructions made using different methodologies but are here plotted together. Reconstructions are plotted relative to recent values, except for the Shakun reconstruction which has been aligned to the Marcott reconstruction for their period of overlap, although such alignment is largely arbitrary.**

Mid-Holocene warmth is also seen in the collection of all calibrated records (Fig. 1), annual-mean records (Fig. 3), and aggregate proxy record statistics (Table 1). It's possible that the proxy database does not give a representative picture of global temperature, which could result from errors in proxy calibrations, errors in the attributed seasonality of records, or a



bias resulting from the spatial non-uniformity of the proxy network. The effect of errors in proxy calibrations is difficult to gauge but, provided that such errors are not too consistent across proxy types, this should be somewhat ameliorated by the diversity of proxy types in the Temperature 12k database. Proxy seasonality was explored in Sect. 3.3, and past work has suggested that their effect should be limited (Kaufman et al., 2020a). As for spatial biases in the proxy network, data assimilation helps account for that directly. However, other uncertainties should be explored, and the uncertainty range displayed for our Holocene reconstruction in Fig. 12 is likely an underestimate. Uncertainties related to proxy record age models, proxy seasonality metadata, and other sources are not represented. Accounting for these areas of uncertainty in the future may help explain the large amounts of spatial diversity even among nearby records in the proxy database (e.g., Fig. 1, Fig. 6).

Some reconstructions have similarities in either the methodology or the underlying data. Our data assimilation approach, for example, uses the same proxy records as the Kaufman composites (with minor updates included in v1.0.2 of the database): we use the Temperature 12k database but omit seasonal records when an annual mean proxy record is available for the same archive. If all eligible proxy records are used instead, the reconstructed climate looks largely the same. The Kaufman composites also use a collection of five different compositing techniques (Kaufman et al., 2020b), all of which differ from the data assimilation method. Like the Kaufman reconstructions, the present Holocene reconstruction shows mid-Holocene warmth, though to a smaller degree.

The present reconstruction uses many of the marine sediments used in the Osman et al., 2021 reconstruction, although we use calibrated versions of these records while Osman uses the raw data together with PSMs. Note that while the Osman et al., 2021 reconstruction shows warming through the Holocene, the source of this apparent Holocene warming remains unclear given that both the underlying marine sediment proxy records (Fig. 7 in Osman et al., 2021) and the mid-Holocene simulations used in the model prior are either warmer or comparable to preindustrial. Without knowing why that reconstruction shows late Holocene warming, it is difficult to explain differences between these two reconstructions.

To compare 40°S-40°N ocean temperature, as used in the Bova reconstruction, to global mean temperature, we calculate both quantities in our new Holocene reconstruction, using air temperatures rather than SSTs. Temperatures over these two domains are highly correlated in our reconstruction, but changes in 40°S-40°N ocean temperatures are only ~57% as large as global mean changes, owing to the large magnitude of temperature changes at higher latitudes (Fig. 5). The use of the 40°S-40°N ocean domain results in a mid-Holocene temperature anomaly of 0.01°C in our main reconstruction.

Recent work has reported improved data assimilation skill by reducing the estimates of proxy uncertainty, which forces the data assimilation to rely more on the proxy information than the prior distribution (Tierney et al., 2020, Osman et al., 2021). As a sensitivity test, we repeat our data assimilation using proxy uncertainty values arbitrarily reduced to 20% of their original values, which is the mean reduction used in past work (Tierney et al., 2020, Osman et al., 2021). In our reduced uncertainty experiment, reconstructed mid-Holocene warmth rises from 0.09°C to 0.17°C, bringing it closer to the Kaufman and Marcott reconstructions but further from the Bova and Osman reconstructions. This is one of many sensitivity





experiments we explore in Appendix B; although the parametric and methodological choices have important impacts, mid-Holocene warmth is a robust feature of our reconstruction.

### 4.4. Caveats and future work

Future improvements in paleoclimate data assimilation may come from a variety of sources. Using a model prior which replicates the climate system's true complexity has the potential to provide the most gains, and improvements in the
global proxy network should also provide clear benefits. Both of these topics are explored in Appendix A. Additional proxy metadata, such as clear indications of whether data points represent contiguous or discrete observations should also aid paleoclimate data assimilation as well as paleoclimate research in general. Such metadata would help researchers understand whether a proxy record with centennial resolution, for example, represents contiguous centennial means as opposed to annual or decadal means sampled at centennial resolution. An extreme data point might represent an important climate event
if it represents a long time period while the same observation may be less remarkable if it only represents a single year.

Additionally, the source of apparently conflicting signals among proxy records must be better understood. Even in well-sampled regions, proxy records present an assortment of diverse signals that cannot all be matched within the data assimilation framework. The sources of these diverse climate signals–whether they result from proxy calibration uncertainties, un-aligned age models, proxy seasonality biases, or something else–is a question for future research.

For shorter-term goals, additional sampling of uncertainties for proxy records (e.g., uncertainties in proxy calibration, age model, seasonality, and more) and model priors (through the use of additional models or alternate prior design) would be beneficial. Additionally, as more proxies are compiled into large, machine-readable databases, Holocene data assimilation can be expanded to reconstruct additional variables such as precipitation. Through future development of the methodology, paleoclimate data assimilation is well-positioned to help scientists infer data about climate fields or regions
where little proxy evidence exists.

### 5. Conclusions

The Temperature 12k proxy database provides considerable information about Holocene temperatures (Kaufman et al., 2020a). Analysis of this database shows general warming in the early Holocene, maximum warmth in the mid-Holocene, and a cooling toward the present day, a pattern which has been shown in past global mean temperature reconstructions
(Kaufman et al., 2020b). To reconstruct spatially complete changes, regions without local proxy data must be inferred based on existing proxy records, which is here accomplished using paleoclimate data assimilation.

This is the first implementation of a multi-timescale paleoclimate data assimilation methodology using real proxy data. By assimilating the data at high temporal resolution using timescale appropriate covariances, we avoid a key assumption required in other approaches, allowing the method to reconstruct high-resolution changes that would otherwise



be obscured. This potential was realized in the reconstruction of a cold anomaly at 8.2 ka, which was reconstructed with spatial and temporal patterns that are generally consistent with previous results.

On longer timescales, the global mean Holocene reconstruction shows peak preindustrial Holocene warmth during the mid-Holocene, consistent with the proxy data. Reconstructed peak mid-Holocene warming was 0.09°C relative to the past millennium, which is cooler than previous reconstructions (Marcott, Kaufman) but warmer than recent reconstructions

that do not simulate a mid-Holocene Thermal Maximum (Bova, Osman). Our assimilation framework also allowed us to test the impact of seasonality explicitly. Summer biases, even when imposed on all records, cannot explain the discrepancy between the proxies and the model simulations. Spatially, the reconstruction shows cold temperatures in regions where the Laurentide and Fennoscandian ice sheets have been reconstructed, adding support for the reconstruction's skill in these well-sampled regions.

By merging paleoclimate data with information from climate models, paleoclimate data assimilation can infer spatially complete climate from incomplete data, a key benefit for exploring past climate. The present paper examines Holocene temperature, but as more proxy data is compiled into large machine readable databases, new long climate simulations are run, and the data assimilation methodology is further refined, this approach is well suited to clarify our perspective on more climate variables and time periods in the past. Reconstructions of past climate help reveal the

characteristics of natural variability, which is the backdrop against which current climate change is rapidly occurring.

**Appendix A: Pseudoproxy tests and proxy/reconstruction agreement**

**A.1. Pseudoproxies tests**

To explore our data assimilation approach, the method is tested using alternate data extracted from a "known" climate. Specifically, temperature data is selected from a variety of locations in a transient model simulation and processed

to create a collection of time series records akin to a proxy network, called "pseudoproxies" (Smerdon, 2011). While pseudoproxies do not contain real data about past climate, they represent a deliberately limited perspective on a known climate, useful for exploring the skill of a reconstruction methodology under controlled conditions. In the primary pseudoproxy experiment conducted in this paper, pseudoproxies use the same locations, seasonalities, and temporal characteristics as the real Temperature 12k proxy records but use temperature values from the closest grid cells of a

Holocene simulation. To account for uncertainty, white noise is generated with a standard deviation equal to the metadata's RMSE uncertainty value for each record. This white noise is added to each pseudoproxy time series after averaging the selected model data into the same temporal windows as the original proxy record.

Several pseudoproxy experiments are run to verify the data assimilation approach. In the primary test case, pseudoproxies are generated from the TraCE-21ka transient simulation and the transient HadCM3 simulation is used as the

prior, ensuring that the pseudoproxies and prior are not derived from the same model data. This differs from the primary data assimilation experiment, where we use both climate models in the prior. Since the TraCE-21ka simulation is used as the





"real" climate, these pseudoproxy experiments test the ability of the data assimilation to reconstruct known climate states in a fashion similar to real reconstructions where the proxies are derived from nature.

The TraCE-21ka transient simulation shows increasing global mean temperature throughout the Holocene. This feature is replicated in the reconstruction (Fig. A1) with a Pearson correlation coefficient of 0.98 between the reconstructed and "true" global mean temperature. Temporal correlations between the reconstruction and model are relatively high across most locations, especially the data-dense regions of Europe and the United States (Fig. A1b). In the most pronounced region of difference—the Southern Ocean off the coast of West Antarctica—the reconstruction produces warmer temperatures in the early Holocene rather than the colder temperatures present in the model. This is one location where the covariance

patterns in the simulations used for the pseudoproxies (TraCE-21ka) and the prior (HadCM3) diverge. In the HadCM3 simulation, this region correlates positively with only ~18% of the rest of the world, while it correlates positively with ~98% of the rest of the world in TraCE-21ka. If the TraCE-21ka simulation is considered the true climate, then the differences between models represents model bias. Without any local data in that region, the reconstructed temperature trend is dictated by these biased covariances. Several other regions of mismatch also stem from differences in the covariance patterns of the

two models.

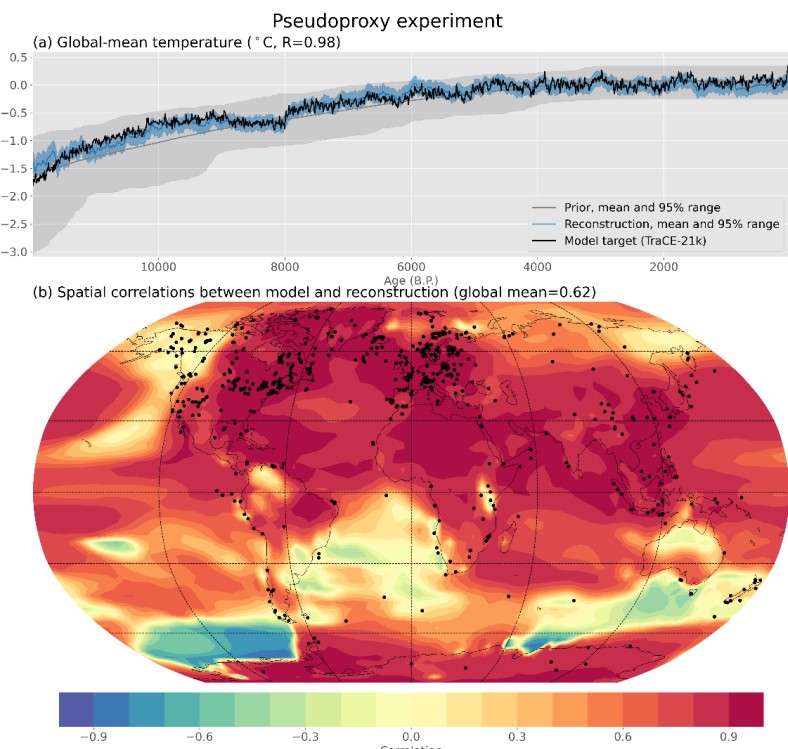

**Figure A1. Temporal and spatial agreement in a pseudoproxy experiment. Temperature pseudoproxies are generated from the TraCE-21ka simulation and reconstructed using data assimilation with HadCM3 as the prior. (a) Annual, global mean temperature in the prior, reconstruction, and model. (b) Correlation between the reconstruction and model temperatures at every location. Locations of pseudoproxies are shown as dots in panel (b) and have the same spatial and temporal coverage as the Temperature 12k proxy database.**





**Figure A2. Spatial skill of pseudoproxy experiments.** Spatial correlations (R) and coefficients of efficiency (CE) for four different pseudoproxy experiments. Global mean values, calculated as the area-weighted mean of the spatial values, as given above each map. Each experiment uses HadCM3 as the prior but differ in the model used to generate the pseudoproxies (Exps. 1, 2: TraCE-21ka; Exps. 3, 4: HadCM3) and the spatial, temporal, and seasonal characteristics of the pseudoproxies (Exps. 1, 3: as in Temperature 12k; Exps. 2, 4: uniform proxy network). When pseudoproxies are based on the Temperature 12k network, they use the spatial, temporal, and seasonal characteristics as the real proxy records ($N_{pseudoproxies}$=711). When pseudoproxies are generated on a uniform 10° by 10° grid, they are all annual mean and cover the entire Holocene with decadal resolution ($N_{pseudoproxies}$=648). More details about these experiments are given in Table 2.



To better understand the reasons behind these mismatches, several more pseudoproxy experiments are conducted. The new experiments implement improvements in two key aspects of the underlying data: a spatially consistent pseudoproxy network (Exp. 2), an "unbiased" prior (Exp. 3), and both (Exp. 4).

For Exp. 2, we generated pseudoproxies from the TraCE-21ka simulation on a 10° by 10° latitude-longitude grid (n=648 pseudoproxies). Seasonal and temporal preferences are also removed, with each pseudoproxy representing decadal climate with no seasonal preference and covering the entire 12 ka time period, with the same amount of noise added to each pseudoproxy. Using this new pseudoproxy network, correlations are slightly improved and the coefficient of efficiency (CE, which is a measure of the fraction of variance captured by the reconstruction, Nash and Sutcliffe, 1970) is greatly improved

in many regions (Fig. A2). These results demonstrate how a proxy network with better spatial coverage and no temporal or seasonal over-representations can improve the reconstruction skill. Regardless, some regions still show errors in the reconstruction. In particular, the Southern Ocean off the coast of West Antarctica still shows negative correlations, suggesting that the presence of local pseudoproxies is not enough to overcome the influence of a large number of remote pseudoproxies with "incorrect" covariances to this region. The influence of long-distance covariances could be diminished or

eliminated through the use of covariance localization, in which the reconstruction is only informed by records within a prescribed radius. Covariance localization has been used in prior work (e.g., Osman et al., 2021; Tierney et al., 2020), and is explored further in Appendix B.

     To test the effect of an unbiased prior, Experiments 3 and 4 use the transient HadCM3 simulation for both the pseudoproxies and the prior. This ensures that the prior covariances match the "true" state covariances, and thus there are no

model biases. Experiment 3 uses the Temperature 12k proxy distribution while experiment 4 uses the uniform proxy network, as in Exp. 2. Both experiments show substantial improvement in correlation and CE values, indicating the importance of an unbiased prior. These experiments show that having an unbiased prior is more important than having a uniformly sampled and seasonally unbiased proxy network (c.f. Exp 3 vs. 2), but the use of both modifications (Exp. 4) produces the best results. The importance of realistic prior covariances has been shown in past work (Dee et al., 2016;

Amrhein et al., 2020).

     Improvements in either the proxy network or model realism should aid future paleoclimate reanalyses. On the topic of model realism, no model is perfect, so we use climate states from two simulations in the main data assimilation experiment to diminish the impact of single-model biases. In the future, the inclusion of more simulations may better emphasize robust multi-model covariance patterns while properly accounting for uncertainty when models disagree, and past

work has supported this approach (Parsons et al., 2021). Further improvements in the proxy network or model realism are beyond the scope of the current work, but will be the natural byproduct of future efforts to improve climate models and proxy databases. Even without such improvements, the relative skill of the pseudoproxy experiment (Fig. A1) supports the use of data assimilation for reconstructing spatial Holocene temperatures, with the caveat that shortcomings in the proxy network and the model prior reduce the accuracy of the results. Note that the prior in each of these experiments is allowed to





change through time, so the prior inherits low-frequency variability from the underlying model. However, if a time-constant

prior is used instead, these general results still hold true.

### Comparison of each proxy and reconstructed proxy, $N_{assim}$=711

**(a) Correlations**

**(b) CE**

**(c) RMSE**

**Figure A3. Agreement between proxy records and reconstructed proxy values. Distributions of (a) Pearson's correlation coefficient, (b) coefficient of efficiency (CE), and (c) root mean square error for each calibrated proxy compared to reconstructed**
**temperature at the same location and season. Comparisons are made on a decadal timescale. Distributions show assimilated records (red) and records which were omitted due to a lack of uncertainty values (n=44, blue). Median values are shown as vertical lines.**





**Figure A4. Comparison of proxy and reconstructed anomalies in space and time. (a) Proxy (symbols) and annual mean reconstructed (background) temperature for the period 6000-6010 vs 3000-5000 years BP. (b) Proxy values vs. reconstructed records for 6000-6010 vs 3000-5000 years BP. (c) The mean of proxy records through time compared to the mean of reconstructed records through time. The (d) slope and (e) correlation between proxy records and reconstructed records through time. Reconstructed records are calculated using the data assimilation method for temperature at the same location and seasonality as the real proxy records.**





**A.2. Proxy records vs. the reconstruction**


To quantify how well the Holocene Reconstruction (discussed in the main paper) agrees with the Temperature 12k proxy database that informs it, we reconstruct temperature time series at the same locations and seasonalities as the original proxy records. These "reconstructed records" are compared to the original proxy time series using three different skill metrics: Pearson's correlation coefficient (R), coefficient of efficiency (CE), and root mean square error (RMSE), calculated

separately for each of the 711 assimilated records as well as 44 un-assimilated records which lacked uncertainty values (Fig. A3). Dissimilarities among nearby proxy records (see Fig. 6) will degrade apparent skill of the data assimilation, as the relatively low-resolution reconstruction will not match such apparent spatial complexity.

Correlation values between the proxies and reconstructed proxies are mostly positive, showing that the general patterns of change are captured, but median CE values are slightly below zero. For CE, values below 0 are generally

considered to represent a lack of skill. If change in skill between the reconstruction and the prior is examined instead, the ΔCE values are slightly positive: 0.15 for assimilated proxies. As stated earlier, the pronounced spatial diversity of the proxy data complicates efforts to match all records simultaneously.

To visualize spatial inconsistencies, the reconstruction and input proxy data are shown for an example decade along with summary metrics plotted through time (Fig. A4). For the chosen decade, proxy data have a much larger range of

anomalies than the reconstructed records, showing that the method cannot match all the records at once and instead finds a middle ground consistent with covariances in the model prior. Consistent with this, the mean of the proxy records matches the mean of the reconstructed records relatively well through time, although the reconstructed proxies have less mid-Holocene to present cooling, likely due to the warming trend in the prior (Fig. 4). The small values of regression slopes indicates that the reconstruction does a poor job matching the spatial diversity of the proxy signals (Fig. A4d). Correlation

values range from ~0.1 to ~0.5 through time, with better correlation in the early Holocene when the climate anomalies are large (Fig. A4e). It is worth noting that these metrics are all calculated for climate anomalies relative to 3-5 ka as opposed to absolute climate values shown in past work (Osman et al., 2021). If the values were calculated for absolute values instead, which include Earth's natural latitudinal temperature gradients, the match would appear far better.

**Appendix B: Alternate experimental designs**

The pseudoproxy tests in Appendix A.1 explored improvements in the proxy network and the accuracy of the model prior. To help account for single model biases, we use two models in the prior and recommend testing the inclusion of additional transient model simulations as they become available. Beyond this, additional improvements in model physics and proxy data acquisition will require considerable future effort and are beyond the scope of this paper. However, other changes can be made to the experimental design. These options are explored in this section, providing a testbed for future

improvements in the data assimilation methodology. In many cases, the philosophy of the current paper was to use the simpler approach for the "default" reconstruction, laying a baseline for future improvements.





We test alternate experimental designs using both real data (Figs. B1, B2) and pseudoproxies (Fig. B3). Five aspects of the experimental design are explored: the use of a constant vs. time-varying prior, the use of covariance localization, the effect of modifying the proxy uncertainty values, the choice of model(s) in the prior, and the use of a 200-year binned proxy approach.

**Figure B1. Holocene reconstruction with different priors. Reconstructions using three different options for prior: (a) the default time-varying prior, which consists of a moving 5010-year window with the mean of 3-5 ka removed, (b) a time-varying prior consisting of a moving 5010-year window with its mean removed at every time step, and (c) a time-constant prior consisting of all climate states centered on 0.5 to 12.5 ka with its mean removed. In all cases, the prior uses climate states from both the HadCM3 and TraCE-21ka transient simulations. Bands represent the 1-sigma (dark shading) and full (light shading) range of the ensemble members. To aid comparison, the mean of the reconstruction in (a) is plotted in black in panels (b) and (c). The reconstruction in panel (a) is the primary reconstruction analyzed in this paper, also shown in Fig. 4.**





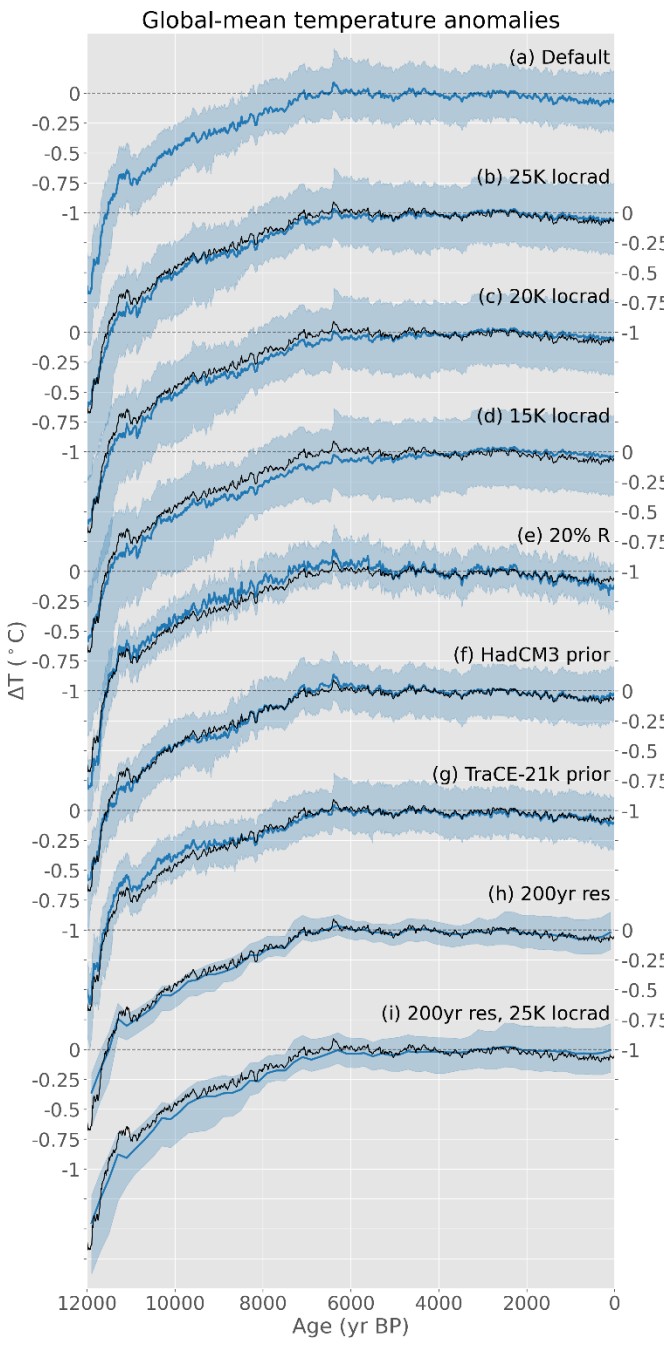

**Figure B2. Global mean temperatures from different experiments. Reconstructed global mean temperature from (a) the default experiment as well as experiments using (b) a 25000 km localization radius, (c) a 20000 km localization radius, (d) a 15000 km localization radius, (e) 20% of the original uncertainty values, (f) the HadCM3 prior, (g) the TraCE-21ka prior, (h) proxies binned to 200-year resolution, and (i) proxies binned to a 200-year resolution as well as a 25000 km localization radius. The mean of the default experiment is plotted in black over the other experiments for comparison. Shading shows the full range of ensemble members for each reconstruction. The reference period of each reconstruction is 3-5 ka. Experimental options are listed in Table 3.**





**Figure B3. Spatial skill of more pseudoproxy experiments. Spatial correlations (r) and coefficients of efficiency (CE) for four**
**additional pseudoproxy experiments, as in Fig. A2. Experiments are the same as the default experiment but use a localization**
**radius and/or a time-varying prior. Exp. 5 uses a time-varying prior 5010 years long, as in the main experiment, but with the mean**
**value set to 0 for every period. Exp. 6 uses a time-constant prior consisting of all climate states centered on 0.5 to 12.5 ka. Exp. 7**
**uses a localization radius of 25,000 km. Exp. 8 uses a time-varying prior of 3010 years, rather than 5010 years in the default**
**experiment. More details about these experiments are given in Table 2.**



| Exp. | Pseudoproxy model | Distribution | Changing prior | Loc. radius | $R_{GMT}$ | $CE_{GMT}$ | $R_{spatial}$ | $CE_{spatial}$ |
|---|---|---|---|---|---|---|---|---|
| 1 | TraCE-21ka | Temp 12k | Yes, 5010yr | None | 0.98 | 0.95 | 0.62 | -0.01 |
| 2 | --- | Basic grid | --- | --- | 0.99 | 0.96 | 0.66 | 0.21 |
| 3 | HadCM3 | --- | --- | --- | 0.99 | 0.98 | 0.84 | 0.73 |
| 4 | HadCM3 | Basic grid | --- | --- | 0.99 | 0.99 | 0.87 | 0.77 |
| 5 | --- | --- | Yes, 5010yr w/ constant mean | --- | 0.97 | 0.70 | 0.47 | 0.02 |
| 6 | --- | --- | No | --- | 0.97 | 0.92 | 0.67 | -0.24 |
| 7 | --- | --- | --- | 25000 km | 0.98 | 0.96 | 0.64 | 0.06 |
| 8 | --- | --- | Yes, 3010yr | --- | 0.98 | 0.95 | 0.64 | -0.08 |

**Table 2. Skill metrics for pseudoproxy tests. Skill metrics for the pseudoproxy experiments shown in Figs. A2 and B3. Metrics are calculated between the reconstruction results and the original model data that the pseudoproxies are built from. Metrics are the correlation (R) and coefficient of efficiency (CE) calculated for either global mean temperature values (GMT) or calculated for temperature at every location and then averaged using an area-weighted mean (spatial). All these experiments use HadCM3 as the prior model. Boxes with dashes indicate a setting is the same as the experimental design of Exp. 1.**






| Exp. | Prior model | Changing prior | Loc. radius | Uncertainty scaling | Proxy resolution | Figs. |
|---|---|---|---|---|---|---|
| Default | HadCM3 and TraCE | Yes, 5010 yr window | None | None | Multi-timescale | 4-12, A3-A4, B1-B2 |
| Constant mean | — | Yes, 5010 yr window w/ constant mean | — | — | — | B1 |
| Constant prior | — | No | — | — | — | B1 |
| 25K locrad | — | — | 25000 km | — | — | B2 |
| 20K locrad | — | — | 20000 km | — | — | B2 |
| 15K locrad | — | — | 15000 km | — | — | B2 |
| 20% R | — | — | — | 20% of default | — | B2 |
| HadCM3 prior | HadCM3 | — | — | — | — | B2 |
| TraCE-21ka prior | TraCE-21ka | — | — | — | — | B2 |
| 200yr res | — | — | — | — | 200 years | B2 |
| 200yr res, 25K locrad | — | — | 25000 km | — | 200 years | B2 |

**Table 3. Experimental design of data assimilation reconstructions. Settings of different reconstructions: the model(s) used in the prior, the time-varying or time-constant construction of the prior, the localization radius, the scaling of the proxy uncertainties, the approach to proxy resolution (multi-timescale or binned), and the figures where each experiment can be seen. Dashes signify**
**values that are the same as the default experiment.**

## B.1. Time-constant vs time-varying prior

When using a time-varying prior, as in this paper, the prior consists of a changing collection of model states to account for slow changes in the mean and covariance patterns of the climate system (e.g., Osman et al., 2021). When using a
time-constant prior, on the other hand, the prior consists of the same model states at every time step, ensuring that all temporal variability is derived from the assimilated proxy records (e.g., Hakim et al., 2016).




To explore the influence of changes in the prior using real proxy data, two new data assimilation experiments are run for comparison with the default experiment (Fig. B1). In the first new experiment, prior climate states are selected from a 5010-year moving window, as in the main experiment, but the mean of the prior ensemble is set to 0 at every time step. This

represents a middle-ground between a time-varying and a time-constant prior, as the covariance patterns can change but the mean state doesn't. In the other experiment, the prior is identical for every time step, consisting of all climate states centered on 0.5 to 12.5 ka.

Compared to the default reconstruction, anomalies in these two new reconstructions are warmer in the early Holocene (Fig. B1). Since data assimilation is a mix of model data and proxy data, it is unsurprising that a warmer prior

would produce a warmer reconstruction during this period. It is especially notable that the reconstruction has positive anomalies between ~6-8 ka in all three cases, providing more evidence for mid-Holocene warmth. Mean mid-Holocene warmth in these three experiments is 0.09, 0.11, and 0.14 °C, respectively. These results demonstrate the potential effects of the prior on the final reconstruction, but also show that the major climate trends are not overly influenced by this choice. If these experimental designs are tested using pseudoproxy data, the time-varying prior generally performs better than either of

these constant-mean experimental designs (compare Exps. 1, 5, and 6 in Table 2 and Fig. A2, B3).

Whether a time-constant or a time-varying prior is used, it is worth considering how the prior influences the final reconstruction (e.g., Fig. B1). The use of a time-varying prior may produce a reconstruction which preferentially resembles prior trends while a time-constant prior may produce a flatter reconstruction. Additionally, while a time-constant prior ensures that all time-varying signals in the reconstruction originate from the proxy data, the lack of information about

changing boundary conditions may bias results. On the other hand, a time-varying prior may limit the size of the prior ensemble, as climate states must be drawn from a moving window rather than from a broader expanse of model output. This last drawback, however, has been mitigated in the current work by the use of multiple models, providing twice the number of climate states to the prior and potentially diminishing single-model biases.

As a final note, if data assimilation is conducted using a time-varying prior, desired analyses should be conducted

on both the prior and the reconstruction to see what information was already present in the model prior. Otherwise, features thought to be based on proxy data may simply originate from the original model simulations. To the degree possible, data assimilation should be conducted multiple times using alternate priors to test the sensitivity of results, as has been done here.

**B.2. Covariance localization**

In this paper's main experiment, all proxy records have the potential to influence the reconstruction across the

Earth, with the length of that influence determined by the climate model's covariance structures. Covariance localization, on the other hand, reduces or eliminates the influence of long-range covariances and forces the reconstruction to rely more on local proxy records. This is done by applying a localization radius such that a given proxy can only influence the reconstruction within a certain distance-weighted area. The length of this localization radius is fundamentally arbitrary, and multiple lengths are generally tested to find a length that minimizes the errors of selected reconstruction criteria (Tardif et





al., 2019, Tierney et al., 2020, Osman et al., 2021). In our pseudoproxy tests, Exp. 7 uses a localization radius of 25000 km, as in the Last Millennium Reanalysis (Tardif et al., 2019) (Fig. B3, Table 2). The localization method uses a Gaspari-Cohn function (Gaspari and Cohn, 1999; Tardif et al., 2019) to reduce the influence of proxy data on locations distant from the proxy itself, reducing to 0 outside of the localization radius, as in the LMR project (Tardif et al., 2019).

When applied to real proxy data, a localization radius of 25000 km produces a climate reconstruction similar to the
main experiment in many ways (Fig. B2). Skill metrics show that the new reconstruction matches proxy records slightly better in some respects, with a median correlation to assimilated proxies of 0.37 rather the 0.35 in the default experiment. Because the use of a localization radius can diminish the influence of proxy data, our reconstructions using a localization radius (Fig. B2b-d) more closely resemble the prior, with slightly cooler mid-Holocene temperatures and larger uncertainty bands.

While the use of a localization radius improves the reconstruction in some regards, the method also poses some challenges. As stated above, a localization radius can diminish the potential impact of proxy data, giving more weight to the temporal evolution of the model prior. Additionally, a localization radius arbitrarily diminishes the influence of long-distance climate relationships which may be valid, instead relying more on individual (potentially noisy) proxies in data poor regions. On the other hand, covariance localization prevents data-rich regions from having an outsized influence on the
global climate reconstruction, which may be beneficial. From a technical perspective, the use of a localization radius necessitates the use of serial proxy assimilation rather than simultaneous proxy assimilation (Whitaker & Hamill, 2002). Serial and simultaneous assimilation approaches produce nearly identical results (for our default experiment, ensemble means are the same and ensemble members differ by no more than 0.05 °C), but simultaneous assimilation is computationally faster, which allows for increased efficiency given limited resources. With that said, the use of a 25000 km
localization radius produces a slight improvement in several of our metrics so, with more testing, it might be a useful change to our experimental design.

### B.3. Proxy uncertainties

In data assimilation, proxy records with larger uncertainties have less impact on the final reconstruction. The Temperature 12k database has uncertainty estimates for each record, but these values are based on proxy type and may not
be accurate. First, these uncertainty values represent uncertainty of absolute temperature values rather than relative values, so they may be too large for our relative temperature reconstruction. On the other hand, it's possible that some aspects of uncertainty were overlooked. Recent work found improved skill by scaling uncertainty values to 20% of their original values on average (Tierney et al., 2020, Osman et al., 2021). To explore the effect of modified uncertainty, MSE values are here similarly reduced to 20% of the original values (Fig. B2e). These reduced uncertainties produce larger temperature
anomalies, with an average mid-Holocene temperature anomaly of 0.17°C as opposed to 0.09°C in the original experiment.

Post-hoc scaling of uncertainty values to improve reconstruction skill has been done in other data assimilation work (Osman et al., 2021; Tierney et al., 2020), but this should be done with care. Ideally, uncertainty values should be record-



specific to account for individual considerations of each record. However, the size of the Temperature 12k database, as well as difficulties in determining record-specific uncertainties, place this beyond the scope of the current work. The use of a

smaller, curated selection of proxy records is another approach but may limit spatial coverage of the data assimilation (King et al., 2021).

### B.4. Choice of model for prior

Another consideration is the use of different model simulations in the prior. We use both the HadCM3 and TraCE-21ka transient simulations in this paper, but sensitivity tests can be run using just one of these models (or other models) as

the prior. The prior influences both the initial range of climate states and the relationships between locations, seasons, and variables, so the choice of model simulation affects how proxy anomalies are translated to the rest of the climate system.

Here, pseudoproxy experiments are conducted using single-model or two-model priors. To avoid giving any experiment an unrealistic advantage, the model used to generate the pseudoproxies in an experiment is never included in the prior. Therefore, to generate independent pseudoproxy data for a two-model prior, we also use a third simulation: the

FAMOUS 10x accelerated transient simulation (Smith & Gregory, 2012). The use of an accelerated timescale may affect prior covariances, so the FAMOUS simulation is not used more broadly in this paper and is only used here out of necessity. In these pseudoproxy experiments, the reconstruction is compared to the "true" climate using several metrics: correlation and coefficient of efficiency of both the global mean temperature and spatial temperatures (Table 4). In these experiments, the HadCM3, TraCE-21ka, and two-model priors all perform relatively well. We use the two-model prior in the main

experiment because the use of multiple models provides the prior with more initial climate states and should diminish single-model biases. Recent work has found that multi-model priors are well-suited to data assimilation (Parsons et al., 2021).

When assimilating real proxy data, global mean temperature reconstructions using the HadCM3 or TraCE-21ka prior share many similarities with the default two-model experimental design (Fig. B2), indicating that global mean temperature is not overly dependent on the particular characteristics of the model prior. As with the other experiments

discussed above (Figs. B1-B3), these experiments touch on areas for potential future improvement in Holocene data assimilation.

### B.5. The use of multi-timescale vs binned data assimilation

In the default experimental design, this paper uses a multi-timescale approach to data assimilation. By using covariances between low- and high-resolution timescales, the method attempts to properly account for the temporal

information of proxies. An alternate approach, which has been used in past work (Osman et al., 2021), is to bin proxy data to a uniform timescale. Since the mean temporal resolution of the Temperature 12k proxy dataset is near 200 years, we bin all proxy data into 200-year intervals, using nearest neighbor interpolation to span intervals between proxy data points. Using this approach, the data assimilation produces a reconstruction that is approximately a smoothed version of the default multi-timescale experiment, with a reduced uncertainty band (Fig. B2). If correlations are calculated for temperature at every





location between this experiment and the default experiment regridded to 200-year resolution, the global-mean of these

correlation values is 0.96. If a 10-year bin is used instead (effectively a single-timescale version of the default experiment),

its mean spatial correlation with the default experiment is also 0.96. Additional comparison metrics should be calculated to

determine the full effects of a multi-timescale approach, which is left to future work.

| Exp. | Pseudoproxy model | Prior model | $R_{GMT}$ | $CE_{GMT}$ | $R_{spatial}$ | $CE_{spatial}$ |
|------|-------------------|-------------|-----------|------------|---------------|----------------|
| 1 | HadCM3 | TraCE | 0.96* | 0.90 | 0.60* | -0.62* |
| 9 | HadCM3 | Famous | 0.93 | 0.87 | 0.51 | -1.08 |
| 10 | HadCM3 | 2-model | 0.96* | 0.92* | 0.58 | -0.84 |
| 11 | TraCE | HadCM3 | 0.98* | 0.95* | 0.62 | -0.01* |
| 12 | TraCE | Famous | 0.95 | 0.89 | 0.57 | -0.57 |
| 13 | TraCE | 2-model | 0.97 | 0.93 | 0.66* | -0.02 |
| 14 | Famous | HadCM3 | 0.94* | 0.86* | 0.45 | 0.14 |
| 15 | Famous | TraCE | 0.94* | 0.84 | 0.47* | 0.15 |
| 16 | Famous | 2-model | 0.94* | 0.86* | 0.46 | 0.17* |


**Table 4. Skill metrics of pseudoproxy tests – choice of prior model. Skill metrics for pseudoproxy experiments, as in Table 2, but exploring the effect of changes in the model used to generate pseudoproxies and the model(s) used in the prior. For the "2-model" experiments, the two models used in the prior are the models not used to construct the pseudoproxies (e.g., if HadCM3 is used to construct the pseudoproxies, TraCE-21ka and Famous are used in the 2-model prior). Asterisks indicate the highest values for**
**each set of pseudoproxies for the precision shown, with ties allowed.**

Code availability: The code to compute the Holocene reconstruction is written in Python and is available at https://github.com/Holocene-Reconstruction/Holocene-code.

Data availability: The complete Holocene reconstruction is available on Zenodo at doi:10.5281/zenodo.6426332.

Author contribution: MPE conducted much of the programming, analysis, and writing, with NS contributing to programming and design of the data assimilation. NPM, NS, SD, and CH contributed to data analysis. RFI, LJG, and PV provided model output. All authors contributed to the writing of the manuscript.


Competing interests: The authors declare that they have no conflict of interest.





Acknowledgements: The authors would like to thank Greg Hakim, Robert Tardif, and the rest of the Last Millennium
Reanalysis (LMR) team for the use of some LMR code in the present data assimilation. The LMR code is available through
https://github.com/modons/LMR.

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
