# Peer review of "Reconstructing Holocene temperatures in time and space using paleoclimate data assimilation"

_EGUsphere, 2022_

## Author Response (AR1)

Review responses for Holocene DA paper

In this document:
- Reviewer comments are shown in blue.
- Our original responses (posted in the preprint discussion) are shown in black.
- Descriptions of the changes made to the paper are shown in bold black.

Reviewer #1:

**Review of the paper by Erb M.P. et al. entitled "Reconstructing Holocene temperatures in time and space using paleoclimate data assimilation"**

General comments

The authors apply paleoclimate data assimilation to create a spatially-complete reconstruction of temperature over the past 12'000 years. They use proxy data from the Temperature 12k database and output from HadCM3 and TraCE-21ka. The high temporal resolution of the analysis allows insights into extreme events such as the 8.2 ka cooling period. Relative to the past millennium the study shows a warm peak near 6'400 years ago, which is with 0.09 °C cooler than in previous reconstructions. This is possibly a more realistic value? The paper is precisely written and convinces with a clear methodological concept.

Thank you for these useful comments. The responses to your comments are below.

Specific comments

-Line 160, data assimilation: An interesting paper was recently published in Climate of the Past by Franke et al. (Clim. Past 16/2020, p. 1061-1074).

- Line 160, data assimilation: Franke et al., 2020 is a good suggestion. We will cite it.

**We have cited Franke et al., 2020**

-Figure 1: I do not understand the values -10 to -70 for the Southern Hemisphere om the right ordinate. Because of the inertia due to the huge ocean bodies, I would have expected a delay of the warmest decade in the Southern Hemisphere compared to the north.

- Figure 1: The figure is showing the calibrated data from the Temperature 12k database. Each record is represented as a colored horizontal line, with records arranged from north to south. The right y-axis helps show the latitude of the records. I will add additional text to the caption to clarify the right y-axis. As for Southern Hemisphere changes, the Southern Hemisphere mid-latitudes arguably do show a later timing of max temperatures than the northern mid-latitudes

(compare the location of the black vertical lines), but it is noisy. This comparison would be aided by a larger network of Southern Hemisphere records.

**The Fig. 1 (now Fig. 2) caption has been slightly edited to clarify the meaning of the right y-axis.**

-Table 1 / temperature trends 6 - 0 ka: The cooling during the LALIA and LIA was mainly dominated by cold winters. The low value of 34.5% is rather surprising.

- Table 1: Most of the proxy records come from the Northern Hemisphere, where winter insolation increased through the late Holocene and summer insolation decreased.  Because of these insolation trends, it is not surprising that more of the annual and summer records show cooling than the winter records.  It's possible, however, that other changes would be seen in a more spatially-complete proxy network.

-Line 271: I would emphasize that the insolation has strongly decreased during the boreal summer.

- Line 271: We will emphasize the decrease in boreal summer insolation. For reference, boreal summer insolation is typically ~450 W/m2 and was ~25 W/m2 greater in the early Holocene, a difference around 5%.

**We have added the word "substantially" to emphasize the insolation change.**

-Lines 326-327: This is another indication of the inertia of the Southern (Ocean-) Hemisphere.

- Lines 326-327: We agree. We will reinforce this point in the sentence.

**On further thought, it seems unlikely the large heat capacity of the Southern Ocean could introduce climate lags on such large timescales (hundreds to thousands of years). Instead, we updated the sentence to mention the relative lack of proxy records in the Southern Hemisphere.**

-Lines 340-342 and lines 535-545: As Figure 4 in Kaufman et al. (Scientific Data 7:115) shows, it is of considerable importance that a distinction is made between different proxy types.

- Lines 340-342 and lines 535-545: That's a good point: differences in proxy types may help explain some of the spatial variability in the Temperature 12k proxy database.  Regardless, Fig. 4 in Kaufman et al. 2020 also shows a fair amount of similarity between composites using different proxy types.

-Line 374: I would emphasise that Holocene insolation was MASSIVELY greater in the boreal summer.

- Line 374: We will emphasize this.

**We have added the word "substantially" to emphasize the change.**

-Comment to lines 411-412 and section 4.3.: The question arises to what extent the temperature increase in the late Holocene simulated by several models can be attributed to the influence of the Southern Hemisphere with its large oceans. Obviously, the number and quality of proxies from this region is insufficient. In general, the number of winter proxies is also very low. This is disturbing because the temperature variability in winter is high.

- Lines 411-412 and section 4.3: The issue of spatial and seasonal biases in the proxy network is an important one. As you mention, Southern Hemisphere and winter records are under-represented in the Temp12k database, and it would be great to have more.  With that said, some reconstruction methods account for these biases: in the data assimilation, spatial covariance patterns are used to help infer Southern Hemisphere temperatures; in Kaufman et al. 2020 ("Holocene global mean surface temperature, a multi-method reconstruction approach"), proxies are composited into latitudinal bands and averaged together using the relative area of each band, which gives the Southern Hemisphere signal the same weight as the Northern Hemisphere signal.  In the data assimilation, covariance patterns are also used to account for proxy seasonality.  More proxies would certainly be helpful, however.  We will add additional discussion of biases to the paper.

**We have added additional discussion of how the method deals with seasonal and spatial characteristics of the proxy data. For seasonality, a paragraph has been added to Section 2.4. For spatial biases, additional text has been added to Section 4.3.**

-Figure 10: The high number of positive anomalies for the period 0-1 ka is rather surprising.

- Figure 10: Yes, the variety of proxy temperature signals at 0-1 ka emphasizes the large amount of spatial variability in the proxy database.

-Line 463: I suggest also to cite more recent papers, e.g. Matero ISO, Gregoire LJ, Ivanovic RF et al., 2017. The 8.2 ka cooling event caused by Laurentide ice saddle collapse. Earth and PlanetaryScience Letters473: 205–214, https://doi.org/10.1016/j.epsl.2017.06.011.

- Line 463: Thank you. We will check to see whether additional new papers should be cited. Matero et al., 2017 is already cited.

**We did not see any additional new papers that needed to be cited.**

Technical corrections

-Citation Osman et al.: The name of the paper (Nature 599, 239-244) is missing.

- Citation Osman et al.: We will fix this citation.

**The citation is fixed.**

Reviewer #2

Erb and colleagues provide a new global temperature reconstruction covering the Holocene by combining available paleoclimate records from the Temp12k database and climate model simulations using a data assimilation method. Such a reconstruction is particularly relevant since proxies only provide information at the local scale and can suffer from seasonal biases. The paper is pleasant to read and most choices are well justified. This study is likely suitable to be published in Climate of the Past after considering my following comments - I have no major comments.

Thank you for the review.  We respond to your comments below.

1. In comparison with Osman et al., continental records are also included in the data assimilation. What is the contribution of these additional records to the reconstruction? In other words, how much of the signal in the global reconstruction comes from 1) ocean records and 2) continental records?

1. Our reconstruction differs from the one presented in Osman et al., 2021 in a variety of ways: the proxy selection, the use of PSMs, and the chosen model prior. Additionally, while we used some of the same proxy records as in Osman et al., 2021, they used additional marine records. The marine sediment records from Jess Tierney that we considered for assimilation were 103 d18O, 63 alkenone, 51 Mg/Ca, and 13 GDGT records, but they have added additional records to their database. We will look into acquiring more records for future work.

With this said, we ran two new experiments to explore the effects of land and ocean proxies on the data assimilation: one experiment only uses land proxy records and the other only uses ocean proxy records. These two experiments share some similarities, but differ in their spatial patterns especially in the early Holocene, where the ocean-only reconstruction is warmer. For the global mean, the default all-proxy experiment has a mean anomaly of 0.09°C for the 6-0.5 ka millenniums (as discussed in the paper) while the anomalies are 0.08°C and 0.04°C for land-only and ocean-only experiments, respectively. The low value of the ocean-only reconstruction suggests that some of the mid-Holocene warmth signal is coming from the land proxies, which are absent from Osman et al., 2021.

2. The Southern Hemisphere contains far fewer records than the Northern Hemisphere. Do you have any clues on the potential impact on global reconstruction?

2. More Southern Hemisphere records would always be welcome, but data assimilation accounts for spatial biases to some extent. Essentially, data in unknown regions is estimated based on known data and model covariances. However, it's likely that the reconstruction would be improved if it had more Southern Hemisphere records to rely on. The localization radius experiments in Fig. B2a-d are somewhat relevant to this question; in the default experiment, the large amount of proxies in the Northern Hemisphere may have an outsized effect on the reconstruction (after being translated through the prior's covariance pattern), but in experiments with a localization radius, proxies cannot influence the reconstruction in distant regions. The

global mean temperature reconstruction is not dramatically different between these experiments.

> 3. The authors chose to work in temperature space by converting all proxies to temperature, but in recent years an increasing number of proxy system models have been published. For example, in Osman et al. such models are used. In my opinion, using more sophisticated PSMs is the way to go to improve reconstructions since proxies are not only temperature-sensitive. I understand that this is a heavy load, but the authors should at least discuss this point – for me, this is the main caveat of the study.

3. There are a few proxies in the network that do have published PSMs, notably ice cores, speleothems, and some of the tree data. However, we decided to use calibrated records for this first paper on the project, and move forward to a sensitivity test using the nonlinear/water-isotope-enabled PSMs for a second study. Furthermore, most of the proxies we could use PSMs for depend on water isotope enabled model output. The availability of water isotope enabled output is also very limited for the Holocene (e.g., iTRACE results have only been released prior to 11 ka and iCESM covers 850-2005), but nevertheless we could use the early Holocene and last2k isotope enabled simulations to conduct sensitivity experiments. The use of such PSMs will be a focus of future work in this project. We will add additional discussion about this to the paper.

**A new paragraph about PSMs has been added to Section 4.4.**

> 4. In the method section, it is said that all the proxies (initially at the decadal resolution) are interpolated annually, then binned into the decadal mean. I don't understand why this is necessary. How does it impact the DA results? For me, this may introduce noise, which could ultimately explain the small variance compared to other DA-based reconstructions (e.g. Osman et al.).

4. The proxy records initially have a variety of temporal resolutions, from sub-annual to multi-centennial. Since we are doing the data assimilation at decadal resolution, we need to have the proxy data at a decadal resolution. As mentioned in the paper, we make the assumption that all proxy data is continuous. To represent longer-time-averaged data in a decadal context, we use a nearest-neighbor approach. Data is first interpolated to an annual resolution using nearest-neighbor interpolation, then binned to decadal. This two-step process is similar to doing a decadal nearest neighbor interpolation, but is meant to better account for sub-decadal data and time intervals mid-way between data points.

If proxy data was instead assumed to represent non-continuous periods (i.e., representing shorter time means with gaps of no data in between), the variability in the reconstruction may be higher, but some of this may be non-climatic variability resulting from records alternating between having data and lacking data. To test this, we ran two new data assimilation experiments. For simplicity, both use 200 year bins. In the first case, data was interpolated. In the second case, no interpolation was done. In both cases, the global-mean temperature reconstruction looked similar. The main difference occurred near 11 ka, where many proxy

records end; in the reconstruction without interpolation, there is greater variability here, but we do not think that such variability represents a real climate signal.

> 5. The model outputs were interpolated on a 2.8125°x3.75° grid. Could you please justify this choice?

5. The two models used in the prior are on different grids, so interpolation is needed to put them on the same grid. The chosen 64 x 96 lat/lon grid is somewhat arbitrary, but is in-between the resolutions of the two models (73 x 96 for HadCM3 and 48 x 96 for TraCE).

> 6. The authors strongly insist on the relevance of the multi-timescale approach in several places. Could you please quantify the added value of using appropriate multi-timescale covariances? I would say that such an approach is needed when performing a data assimilation experiment with annual and decadal (or even lower) records to keep the memory of the ocean, but I am wondering if this multi-timescale approach is really needed here because you have assimilated 10-year and lower resolution proxies.

6. There are multiple reasons why one would use a multi-scale approach and as you say, one reason would be to capture longer term ocean memory or dynamics. For our reconstruction using multiscale assimilation is particularly important because we have proxy data on very different timescales, from multi-centennial sediment records to high resolution ice core records. Given timescales that differ by orders of magnitude, we wanted to treat each proxy on its own timescale and not assume that all proxies provide meaningful information on only a single timescale. It is a straightforward exercise to show that averaging or interpolating high resolution data down to low resolution removes high frequency variability and spectral information; this is precisely what happens when we assimilate proxy data that has been bin-averaged: high-frequency information is lost in the reconstruction. Additionally, there will be differences in the covariance structures in the prior at decadal vs. multi-centennial scales. We will revise the main text to discuss this issue more fully.

**We have added additional discussion of the multi-timescale approach to the end of Section 2.4.**

> 7. As noted by the authors, annual reconstructions could be biased due to seasonality. In the method section, the authors should explain better how they deal with this to finally produce an annual reconstruction – there are a few words in the results section but it should be clearly mentioned in the method section and in more detail considering its crucial importance.

7. We will add additional explanation of how the data assimilation methodology handles seasonal proxy records.

**A paragraph about this topic has been added to Section 2.4.**

> 8. In the discussion section, the authors mention that the marine sediments used in Osman et al. are also used in their reconstruction. To better understand why there are

differences between your reconstruction and Osman et al., assimilating only those records into your data assimilation scheme would be very useful (e.g. the two reconstructions are based on different methods, albeit similar). It would also be an opportunity to quantify the contribution of marine archives in the final reconstruction (see my previous comment), and to quantify the use of PSMs.

8. As mentioned in comment 1 above, Osman includes marine records which were not included in our reconstruction. Also, because the two methods differ in their prior and use of PSMs, a straightforward comparison is not possible. The comparison between land-only and ocean-only proxies described in comment 1 above partly addresses this topic.

9. Proxy uncertainty plays a major role in the final DA-based reconstruction because it determines the weight given to each proxy record in data assimilation. The lower the proxy uncertainty, the higher its contribution. The authors performed an additional reconstruction by decreasing the proxy uncertainty by 20%. In order to know if the proxy uncertainty has been correctly estimated, some indicators on the size of the ensemble exist, for example, ECR or CRPSS. Quantifying the ensemble size for all analyzed reconstructions could help to better understand the differences between the reconstructions and may explain why your reconstruction has a reduced variance compared to Osman et al. reconstruction.

9. To look into this, we did a test similar to the one on Extended Data Table 1 of Tierney et al., 2020 ("Glacial cooling and climate sensitivity revisited"). A subset of proxies was withheld from the data assimilation, then correlation, CE, and RMSE values were calculated for un-assimilated proxies at each age. As in Tierney et al., proxy scaling factors of 1 (the default), 1/2, 1/5, 1/10, 1/20, 1/100, and 1/500 were tested. Median verification metrics were best for values of 1, 1/2, and 1/5. Since none of the other scaling factors were unequivocally better than the default, we will continue to use the default values. However, a reconstruction using a 1/5 scaling factor is still shown in Fig. B2 of the paper. As for ECR and ECPSS, we are unfamiliar with those methods. If you provide further explanation for how those methods would help, we can investigate their use.

10. Although somewhat out of scope for this study, I encourage the authors to also provide atmospheric circulation reconstruction (e.g. sea level pressure, 500-hPa geopotential heights).

10. Reconstructing additional climate variables will be a focus of future work. Before providing reconstructions of those variables, we would like to assimilate more proxy records. In particular, we are currently working to assimilate hydroclimate proxies.

Community comment #1:

Erb et al conduct a new DA of the Holocene using the Temp12K database. As an advance over previous work (i.e. Osman et al), they include terrestrial temperature data. Generally they find a fairly flat temperature trajectory across the Holocene, with less overall variance in global T than previous reconstructions.

I think this is a useful contribution and it's interesting to compare/contrast this result with Osman et al., 2021 in particular. However it remains unclear to me why the Erb et al reconstruction has less variance than previous work. It would be helpful if the authors could identify why this is the case. I wonder whether it has to do with the technical choices made in the DA. Although the testing in the Appendices suggests that the same result is coming out under different experiment assumptions, I wonder if perhaps the use of interpolation on the proxy data is playing any role (see 2. below).

Thank you for these comments. We've replied to your comments in order below.

Here are some general comments:

1) I am not clear on why the authors chose decadal as their base resolution. The mean temporal resolution of Temp12K is 200 years, as stated in Appendix B.5. This is why, in Osman et al., we chose 200 years as our reconstruction bin size (90% of the data have this resolution or better). Perhaps some of the terrestrial data include finer-scale records but for lakes, even if the sampling resolution is decadal is the lake is not varved there is no way that represents the actual age resolution because bioturbation would smooth the signal. Is there really any recoverable information in the proxy database below 200 years? The authors need to justify their 10-yr choice, especially given that Appendix B shows that 200-yr bins give essentially the same result. I'm just concerned that users of this DA will think that the decadal resolution of this product is "real"... when effectively it is not given that most of the underlying data does not have information at this timescale.

1) Proxy records in the Temp12k database have a large range of temporal resolutions. Some ice core, speleothem, and two wood records have high resolution, although these archive types are in the minority. Additionally, a considerable amount of marine sediment, lake sediment, and peat records have mean resolutions lower than 200 years. While bioturbation may smooth some of this signal, our approach attempts to retain higher resolution information rather than smoothing the proxy signals up front. This results in a climate reconstruction which, while nominally decadal, simply represents the information contained in the proxy database. We will clarify this in the paper and stress that the decadal nature of the reconstruction does not imply that it contains robust decadal information. Instead, we want to use the data present in the database. This is similar to the approach used in Marcott et al, 2013 ("A Reconstruction of Regional and Global Temperature for the Past 11,300 Years"), which presents a temperature composite with a resolution of 20 years despite a median resolution of 120 years in their proxy data.

**We have added text to Section 2.2 to stress that, while the reconstruction is nominally decadal, its information content is dependent on the assimilated proxies.**

2) The proxy data information is interpolated and I wonder if this impacts the DA results. I'm reading correctly, it seems like the data are interpolated annual resolution then re-binned to 10-years in all cases? (Line 206)? Any sort of interpolation in my view is unideal, because you effectively introduce imaginary proxy information (that assumes the climate trajectory between data points is linear). Moreover I see that Line 224 indicates that proxy data is used repeatedly for all time steps that it spans. I think this could potentially contribute to the relatively flat posterior that the authors get, although I'm not sure.

Binning is a more robust approach because it ensures the proxy information is only used in one time interval (hence binning has been used for many of the PAGES reconstructions). I see that binning was testing in Appendix B.5, but it looks like the proxy information was still interpolated in this case, hence the information is still used in multiple bins. Instead, put the proxy data into only one bin, but then resample age models and thus shuffle proxies between the time-bins (as we did in Osman et al) to account for the fact that you don't know exactly the age of (most) of the data. That would be a better test and would also assess whether the interpolating is biasing the result at all.

2) In our approach, the data assimilation is computed on a decadal timescale, so we need to have assimilated proxy values on the same timescale. To regrid proxy data to a decadal resolution, nearest neighbor interpolation was used to regrid data to annual resolution (e.g., the value for a given year is equal to the closest value), then binned to decadal. The intermediate step of interpolating to annual is meant to better account for sub-decadal data and decades mid-way between data points. A similar approach is used in some of the composites in Kaufman et al., 2020 ("Holocene global mean surface temperature, a multi-method reconstruction approach"). The approach is simply meant to represent multi-decadal data on a decadal timescale. With this said, we may take a different approach that also addresses age uncertainty concerns; see #3 below for more explanation.

To investigate how our interpolation approach affects the reconstruction, two reconstructions were compared. Both use 200 year bins. In one case, proxy data was interpolated between bins using nearest neighbor interpolation (this reconstruction is shown in Fig. B2h of the paper). In the other reconstruction, no interpolation was done. Both approaches lead to similar global-mean temperatures. The uncertainty range of the reconstruction using no interpolation was larger, but only by a small amount. The main difference between the reconstructions occurs near 11 ka, where many of the proxy records are cut off. The non-interpolated reconstruction has a rapid change in temperature here, which is likely not a climate signal, while the interpolated version is smoother.

**We have added some additional text to clarify the temporal regridding process to Section 2.4.**

3) Age model uncertainty. I don't think it is accounted for? I should be included as all the marine and lake records (except varved lakes perhaps) will be radiocarbon dated with uncertainties of at least several decades. You can simply do ensemble DA experiments to sample the uncertainty (this is what was done in Osman et al.). Age model uncertainty is another reason why a 10-yr bin is too optimistic IMO.

3) Currently, age model uncertainty is not accounted for in the reconstruction, as mentioned in lines 542-543 & 586 of the paper. To account for age model uncertainties, we plan to rerun the reconstructions using a new methodology, which will work as follows: for each proxy record, we will generate a set of possible proxy realizations using the geoChronR R package (https://nickmckay.github.io/GeoChronR/) that sample the proxy's age and magnitude uncertainties. Each proxy's ensemble will be used to quantify the median proxy estimate and the joint age/magnitude uncertainty at a decadal timescale. (This approach will be used instead of the nearest neighbor interpolation approach described in the last comment.) The median proxy estimate will be assimilated and the joint age/magnitude uncertainty will be used to define a time-varying uncertainty term at each decade for each proxy (i.e., the R term in Eq. 2 of the paper). By combining age uncertainty and magnitude uncertainty into the R term, this method will account for age uncertainty without having to run a large collection of reconstructions. To our knowledge, this implementation hasn't been used yet for paleoclimate DA, but should account for proxy uncertainties with a reduced computational expense. If there is a good reason why this is not a good solution, please let us know.

**The new method to account for age uncertainty (described above) has turned out to be more work than expected. Additionally, it is untested, so its potential benefits are unknown. As such, we decided not to implement this change in the current paper. Instead, we plan to experiment with it more in the future to determine how/if it is a good approach. We have added some text to Section 4.4 to mention the potential effects of accounting for age uncertainty, and that we plan to explore it more in future work.**

4) Is the multi-timescale DA approach needed? As I read it this simply uses a different covariance based on each proxies' resolution, but I'm not clear how it is appropriate to do that when your base time step is always 10 years. If that's the case, then it seems to me that a 10-yr covariance should always be used. Regardless, I'm not convinced it is actually needed because, model covariance 10-years and longer tends to be very similar (see the experiments I did in my 2020 LGM paper testing the DA with different time-averages). When most of your proxies are low-res, you might as well use just a standard time for both the assimilation and the covariance.

Accordingly, Appendix B.5 suggests that the multi-timescale method isn't adding much. So there needs to be some justification for using it - what is the added value?

4) In the multi-timescale DA approach, the model data is used to compute covariances between different temporal resolutions. To use an example, let's say that a proxy data point represents a 50 year mean. To relate 50 year means to the decade being reconstructed, the model-based proxy estimate is computed as a set of 50 year means, with each 50 year mean centered (to the

extent possible) on the decades in the prior using a running average. In theory, this lets us quantify how 50 year means relate to decadal climate, and this information is used in the data assimilation. Additionally, since a data point representing a 50 year mean spans 5 decades, the process is repeated for each of the decades, with the only difference being our use of slightly different model values due to the moving prior window. The use of a running mean for multi-decadal averages is not perfect, but difficult to avoid without having a very large number of modeled years with which to construct the prior. In the paper, we will clarify the description of the multi-timescale data assimilation approach.

This multi-timescale approach is theoretically useful, as it is designed to allow each proxy to inform the reconstruction on its own timescale. While the final result is not drastically different from a reconstruction made using a single-timescale binned approach, this paper is partly an exploration of new methodological choices. We feel like that is a good justification for using the method and alternate experimental designs are shown in Appendix B. Regarding model covariances on different timescales, it's possible that the models do not properly quantify climate covariances on different timescales; however, that would be a shortcoming of the data rather than the method. In the released code, if users do not want to use the multi-timescale approach, it can be disabled by setting two variables (time_resolution and maximum_resolution) equal to each other in the config file.

**We have added additional discussion of the multi-timescale methodology to the end of Section 2.4.**

5) Regarding the paper's claim of the DA having mid-Holocene "warmth": I wouldn't call an anomaly of 0.09C as evidence of warmth. seems like a stretch - and not significant within uncertainties. Considering the uncertainties, this DA shows a flat Holocene trajectory since 7 ka and that's how the results should be described (unless 0.09 can be proven to be statistically significant).

5) You're correct that 0.09C may be too small to be considered "warmth," as suggested by the uncertainty bands in Fig. 12, which include 0 from ~8 ka to the present. Using a t-test of two related samples of 1002 values each (the size of the prior), the mid-Holocene period (5.5 - 6.5 ka) is significantly different from the recent period (0-1 ka). However, it's unclear if this is the right significance test to use. If ensemble members are analyzed individually, 88% of them have a positive anomaly at the mid-Holocene and 12% have a negative anomaly. We welcome better suggestions for a significance test. If a clear significance test cannot be established, the paper text can be modified to remove the assertion of mid-Holocene warmth, instead stating that our reconstruction does not support a cooler mid-Holocene.

**We have added additional discussion of ensemble members (as discussed above) to Section 4.3 and have rephased a sentence in the abstract. While it is still not clear what the right significance test for this is, we continue to use the word warmth at some points in the paper to indicate that the reconstruction shows maximum mean temperature anomalies during the mid-Holocene.**

6) Finally, in addition to the lack of incorporation of age uncertainty (which I think can and should be addressed) the authors should be clear in the manuscript that a major caveat of their DA is that they are not using proxy forward models. These models are available and out there, and indeed we used the marine ones in Osman et al. I understand that it is easier to work in temperature space but the reality is that many of these proxies (pollen, marine Mg/Ca and d18O, ice core isotopes) are multivariate and not exclusively sensitive to temperature and working in T space ignores these aspects of the proxies.

6) For this first paper, we use the calibrated data from the Temperature 12k database, which is readily available for a wide range of proxy types. PSMs provide many potential advantages for data assimilation and represent an exciting avenue for exploration, but are not available for all of the proxies we use (e.g. pollon). Additionally, some PSMs are best used with isotope-enabled model simulations, which limits the choices of prior for the Holocene. The use of PSMs will be a focus of future study.

**We have added a new paragraph about PSMs to Section 4.4.**

Here are some specific comments:

Line 65: I would be more specific and say the main difference is the inclusion of terrestrial proxy records.

Line 65: We will be more specific about the inclusion of land records. However, it is difficult to describe the full difference in the databases, since Osman also includes many marine sediment records which are not currently in the Temp12k database.

**We have added a sentence about this to the end of Section 1.**

Line 80: reiterate what the age control and time-resolution criteria are for inclusion

Line 80: We will clarify the time-resolution and age control proxy criteria.

**We have added text about this to the first paragraph in Section 2.1.**

Line 98: since this is relevant for your choice of base resolution (decadal) show a histogram (like ED Fig. 1 in Osman et al) of the proxy time resolution.

Line 98: We will include a figure of proxy temporal resolution.

**A new figure (new Fig. 1) has been added to show the median age resolutions of the different archive types used over the period 0 – 12 ka. To account for this new figure, the numbering of other figures has been updated. Additionally, we now describe median proxy resolution instead of mean proxy resolution at several points in the paper.**

Line 99: this isn't a good assumption -- it is probably not true for most of the data. Researchers typically sample at discrete depth intervals (e.g., 0, 4, 8, 12 cm) and do not sample cores

contiguously. You could check this by investigating the depth sampling intervals in Temp12K. This is another reason that arguably the best approach to doing DA on this timescale is to bin the data rather than interpolate.

Line 99: It would be useful to have a better sense of the temporal coverage of data points. However, that data is not widely available in the Temp12k database. As such, we've decided to treat data as continuous and mentioned our justification in the paper. While unrealistic, this assumption has the benefit of avoiding possible non-climatic signals generated by the alternation between proxy data and data gaps. To explore the effect of interpolating proxy data, we did the comparison described in point #2 above; however, we plan to replace the interpolation approach with the ensemble approach described in point #3 above.

Line 126: "minor checkerboard"?? That sounds like a modeling artifact. Can you elaborate, and would this affect the DA? Introducing spatial model artifacts would certainly mess with the covariance.

Line 126: The minor checkerboard pattern is present in the results of the HadCM3 simulation. It seems to result from processes involving the ocean streamfunction and the model orography, although the details of this are not completely clear. In time-slice simulations, rerunning a simulation with smoothed orography causes the checkerboard pattern to disappear but the general climate remains the same. Because of this, we decided to simply apply a smoothing filter to the model results, which is mentioned in line 126 of the paper. We do not expect this to have much effect on the reconstruction.

Line 137: If I'm reading this correctly, you are stating that decadal resolution is equal to or higher than the mean resolution of most of the proxy data? If that's the case, then decadal is not the right choice.

Line 137: Yes, decadal resolution is higher than the mean resolution of most of the records. This is done by design, because we want to utilize information from our high-resolution records. Making use of data at a variety of temporal resolutions is one of the primary goals of the multi-timescale data assimilation methodology. As mentioned in point #1 above, the widely-cited Marcott 2013 proxy composite also uses a temporal resolution at the high end of the utilized proxy data. Regardless, we will add a note that our reconstruction is "nominally" decadal, but actually represents the information content in the proxies themselves.

**We have added text to Section 2.2 to mention that the reconstruction is "nominally" decadal.**

Line 158: Lack of robust PSMs? This is offensive to those of us who have spent a large portion of our research career developing forward models for temperature proxies. It's not just me either. The Mg/Ca community has made major strides in this, as has the pollen community (see for example Parnell et al., 2016, QSR: https://doi.org/10.1016/j.quascirev.2016.09.007), as has your co-author Sylvia Dee. You didn't do the DA in proxy space because it is was easier to do it

in temperature. That's fine, but this is a caveat of your DA, so just admit that rather than erroneously claiming that the forward models weren't there (they are).

Line 158: We agree this wording was unfortunate, since of course there are many PSMs that have been published by both you and by Sylvia and others. We will rephrase this statement. This first study uses the calibrated temperature records and we plan to move forward with exploration of proxy system models in the future. One complication to the use of proxy system models is that many use isotopes as inputs, and there is a limited availability of isotope-enabled simulations relevant to the full Holocene. Osman et al. 2021 used newly-run isotope-enabled simulations for their prior, but running new simulations is a considerable undertaking. Instead, we plan to explore the use of existing simulations, such as iTRACE. Additionally, by using calibrated temperatures for this first reconstruction, we can use it as a baseline to explore the added value of proxy system models in the future.

**We have removed the phrase mentioning the "lack of robust PSMs". Also, a new paragraph has been added to Section 4.4 about PSMs.**

Line 515: mid-Holocene anomalies are usually calculated across some kind of time window (i.e., 5-7 ka minus 0-2 ka). Here is says 6 - 0.5 but is there a window around those centered values? Presumably yes, since mid-Holocene is not a single instance in time. Up to you what window to use but since the other reconstructions are lower res I would do at a minimum 5.5-6.5 (1000 years) minus 0-1 ka or something like that.

Line 515: This calculation uses the time windows of 5.5 - 6.5 ka vs. 0 - 1 ka. This is specified on line 515.

Line 518: Is the 0.09 "warmth" statistically significant when compared to the baseline (presumably, last 2K)? It doesn't really look like it is based on Figure 12. Or else marginally. Provide a p-value. Also I would calculate this over a larger window as suggested in the previous comment.

Line 518: This is discussed in comment #5 above.

Line 537: Reference here some of the work we did in Osman et al to try and get at the origin of this difference.

Line 537: We will cite Osman et al., 2021 in this discussion.

**Osman et al., 2021 has now been cited in Section 4.3.**

Line 576: To some extent you have this metadata in the form of the depth intervals that researchers averaged over for their analyses (in the ideal case, researchers should list both the starting and ending depth of their data point, but they do not always do this).

Line 576: Unfortunately, metadata about sample thickness is not present in the Temp12k database for the vast majority of records. It would be good to have this information standardized and included in future databases.

General comment on localization (Appendix B.2): The results here are consistent with what we have found, which is that some localization improves the reconstruction. While I understand the argument not to localize, on the flip side, localization was designed in the first place to eliminate spurious covariance structures. So why not use some localization in your reconstruction? I think you need to justify, in the main text, why localization is not used.

Regarding localization: Covariance localization presents some benefits and some drawbacks, which are discussed in Appendix B.2. Among the drawbacks: 1) it is not clear which long-distance relationships are valid, so covariance localization is largely arbitrary and 2) covariance localization diminishes the overall impact of proxies on the reconstruction, allowing the prior to have greater influence on the reconstruction. However, the approach also has benefits, so we show some experiments using covariance localization in Fig. B2 and localization is an option in our released code. We will continue to explore its use in future reconstructions, and will clarify the discussion in Appendix B.2 if needed.

**We have revised the text in Appendix B.2 to clarify our decision not to use covariance localization in the main reconstruction.**

Line 835: This is incorrect. We have implemented localization in the joint update and that is what is used in both Tierney 2020 and Osman 2021. c.f. the DASH codebase on GitHub if you want to investigate the method.

Line 835: Interesting! I didn't realize that the covariance localization was possible with the simultaneous data assimilation approach. We will fix this oversight in the text.

**We have removed the text in question.**

**Other minor edits have also been made.**